# Rhomboid intramembrane protease YqgP licenses bacterial membrane protein quality control as adaptor of FtsH AAA protease

Jakub Began[1,2,†], Baptiste Cordier[3,†], Jana Březinová[1,4], Jordan Delisle[3], Rozálie Hexnerová[1], Pavel Srb[1], Petra Rampírová[1], Milan Kožíšek[1], Mathieu Baudet[5], Yohann Couté[5], Anne Galinier[3], Václav Veverka[1,6], Thierry Doan[3,7,*] & Kvido Strisovsky[1,**]

## Abstract

Magnesium homeostasis is essential for life and depends on magnesium transporters, whose activity and ion selectivity need to be tightly controlled. Rhomboid intramembrane proteases pervade the prokaryotic kingdom, but their functions are largely elusive. Using proteomics, we find that *Bacillus subtilis* rhomboid protease YqgP interacts with the membrane-bound ATP-dependent processive metalloprotease FtsH and cleaves MgtE, the major high-affinity magnesium transporter in *B. subtilis*. MgtE cleavage by YqgP is potentiated in conditions of low magnesium and high manganese or zinc, thereby protecting *B. subtilis* from $Mn^{2+}/Zn^{2+}$ toxicity. The N-terminal cytosolic domain of YqgP binds $Mn^{2+}$ and $Zn^{2+}$ ions and facilitates MgtE cleavage. Independently of its intrinsic protease activity, YqgP acts as a substrate adaptor for FtsH, a function that is necessary for degradation of MgtE. YqgP thus unites protease and pseudoprotease function, hinting at the evolutionary origin of rhomboid pseudoproteases such as Derlins that are intimately involved in eukaryotic ER-associated degradation (ERAD). Conceptually, the YqgP-FtsH system we describe here is analogous to a primordial form of "ERAD" in bacteria and exemplifies an ancestral function of rhomboid-superfamily proteins.

**Keywords** ER-associated degradation; intramembrane protease; membrane transporter; proteostasis; rhomboid

**Subject Categories** Microbiology, Virology & Host Pathogen Interaction; Post-translational Modifications & Proteolysis; Translation & Protein Quality

**The EMBO Journal (2020) 39: e102935**

See also: **G Liu et al** (May 2020) and **JD Knopf & MK Lemberg** (May 2020)

## Introduction

Rhomboids are intramembrane serine proteases widespread across the tree of life. In eukaryotes, rhomboid proteases regulate signalling via the epidermal growth factor receptor, mitochondrial quality control or invasion of malaria parasites into the host cells (reviewed in Urban, 2016). Rhomboid proteases also pervade the prokaryotic kingdom (Koonin *et al*, 2003; Kinch & Grishin, 2013), suggesting sufficient biological significance for evolutionary conservation and expansion. However, their functions in bacteria are much less well-understood than in eukaryotes. Notably, in eukaryotes, the rhomboid superfamily includes a number of pseudoproteases of important biological roles, such as iRhoms and Derlins, but the mechanistic and evolutionary aspects of rhomboid pseudoprotease functions have not been clarified (Ticha *et al*, 2018).

In the Gram-negative eubacterium *Providencia stuartii*, the rhomboid protease AarA is required to process a pro-form of TatA, a component of the twin-arginine translocase secretion apparatus (Stevenson *et al*, 2007), but this function does not seem to be widely conserved because most bacteria encode a mature form of TatA. In the archaebacterium *Haloferax volcanii*, a rhomboid protease is involved in protein glycosylation of the S-layer (Parente *et al*, 2014; Costa *et al*, 2018), but the molecular mechanism has not been uncovered yet. Similarly, the main model rhomboid protease GlpG of *Escherichia coli*, widely distributed in Gram-negative bacteria, is extremely well-characterised structurally and mechanistically (Strisovsky *et al*, 2009; Baker & Urban, 2012; Zoll

1 Institute of Organic Chemistry and Biochemistry, Czech Academy of Science, Prague, Czech Republic
2 Department of Genetics and Microbiology, Faculty of Science, Charles University, Prague, Czech Republic
3 Laboratoire de Chimie Bactérienne (LCB), Institut de Microbiologie de la Méditerranée (IMM), CNRS, UMR 7283, Aix Marseille Univ, Marseille Cedex 20, France
4 Department of Biochemistry, Faculty of Science, Charles University, Prague, Czech Republic
5 CEA, Inserm, IRIG-BGE, Univ. Grenoble Alpes, Grenoble, France
6 Department of Cell Biology, Faculty of Science, Charles University, Prague, Czech Republic
7 Laboratoire d'Ingénierie des Systèmes Macromoléculaires (LISM), Institut de Microbiologie de la Méditerranée (IMM), CNRS, UMR 7255, Aix Marseille Univ, Marseille Cedex 20, France
 *Corresponding author. Tel: +33 4 91 16 44 85; E-mail: tdoan@imm.cnrs.fr
 **Corresponding author. Tel: +420 2201 83468; E-mail: kvido.strisovsky@uochb.cas.cz
 †These authors contributed equally to this work

*et al*, 2014), yet its biological function is unknown. A recent report proposed a role of GlpG in the survival of a pathogenic strain of *E. coli* in the mouse gut (Russell *et al*, 2017), but the mechanism has not been demonstrated and it is not clear how specific the phenotype is, because rescue experiment has not been reported. Current knowledge of functions of rhomboid proteases in Gram-positives is similarly sketchy. An isolated report indicated that YqgP may play a role in cell division and may be required for glucose export (hence the therein coined alternative name GluP) (Mesak *et al*, 2004), but rescue experiment was not reported and the involved substrates were not investigated, leaving the molecular mechanisms of these functions unknown.

Therefore, to shed light on one of the significantly populated subclasses of bacterial rhomboid proteases in a model Gram-positive organism, we focused on *Bacillus subtilis* YqgP and used quantitative proteomics to identify its substrates and interactors. We find that YqgP cleaves the high-affinity magnesium transporter MgtE and that YqgP interacts with the membrane-anchored metalloprotease FtsH. At low extracellular concentration of magnesium cations and high concentration of manganese or zinc cations, cleavage of MgtE by YqgP is potentiated, and the globular N-terminal cytosolic domain of YqgP represents the manganese/zinc-sensing unit. The second molecular role of YqgP is presenting MgtE or its cleavage products as substrates to FtsH, for which the proteolytic activity of YqgP is dispensable but its unoccupied active site is essential. YqgP thus fulfils both protease and pseudoprotease functions in tandem with FtsH, representing an ancestral proteolytic platform dedicated to the regulated degradation of polytopic membrane proteins, functionally equivalent to regulatory ERAD in eukaryotes. Our results shed light on the evolution of membrane proteostasis control in response to environmental stimuli, and on the arising of pseudoproteases, which are surprisingly common in the rhomboid superfamily (Adrain & Freeman, 2012; Freeman, 2014).

# Results

### Quantitative proteomics reveals candidate interactors and substrates of *Bacillus subtilis* rhomboid protease YqgP

*Bacillus subtilis* genome encodes two rhomboid protease genes, *ydcA* and *yqgP* [also known as *gluP* (Mesak *et al*, 2004)]. No proteolytic activity has been detected for YdcA so far (Urban *et al*, 2002b; Lemberg *et al*, 2005), while YqgP is a commonly used model rhomboid protease cleaving a number of synthetic substrates (Lemberg *et al*, 2005; Urban & Wolfe, 2005; Ticha *et al*, 2017a,b). YqgP has homologs in a number of Gram-positive bacteria, including *Bacilli,* but also *Staphylococci, Listeriae* and others (http://www.ebi.ac.uk/interpro/protein/P54493/similar-proteins), and thus represents an attractive system to explore the cell biology and functions of bacterial rhomboid proteases. To reveal the repertoire of YqgP substrates, we generated a strain deficient in *yqgP* and re-expressed *yqgP* or its catalytically dead mutant S288A (Lemberg *et al*, 2005) in this background from an ectopic chromosomal locus (strains BS50 and BS51, respectively, see Table EV2). We then conducted quantitative proteomic analysis of membrane fractions of both of these strains employing SILAC labelling and SDS–PAGE pre-fractionation (GeLC

experiment), where we were seeking peptides belonging to proteins migrating at a lower-than-expected apparent molecular weight selectively in the YqgP-expressing strain relative to the S288A mutant expressing one. This analysis yielded the high-affinity magnesium transporter MgtE as the top candidate substrate of YqgP (Fig 1A–C, Dataset EV1, PRIDE dataset PXD014578).

In a complementary approach, we used affinity co-immunopurification and label-free quantitative proteomics to identify proteins associating with YqgP (Fig 1D and E). We ectopically expressed either a functional YqgP-sfGFP fusion or catalytically dead YqgP.S288A-sfGFP fusion as the sole copy of YqgP in the cell. We solubilised the isolated membranes using the NP-40 detergent, isolated YqgP-sfGFP by anti-GFP affinity pull-down and analysed the co-isolated proteins by MS-based proteomics. Using this approach, we identified several high-confidence interactor candidates including the membrane-anchored protease FtsH, and ATPase subunits A, D, F and G (Fig 1E, Dataset EV2, PRIDE dataset PXD014566). The only high-confidence overlap between the two proteomic datasets was the high-affinity magnesium transporter MgtE (Fig 1C and E), promoting it to the highest likelihood candidate substrate. MgtE is the main magnesium transporter in *B. subtilis*, it is essential for magnesium homeostasis (Wakeman *et al*, 2014), and we thus examined the possible functional relationship between YqgP and MgtE.

### YqgP cleaves the high-affinity magnesium transporter MgtE between its first and second transmembrane helices

To validate the results of quantitative proteomics and gain more insight into the role of YqgP in the physiology of *B. subtilis*, we first examined the proteolytic status of endogenous MgtE by immunoblotting using an in-house generated antibody recognising its N-terminal cytosolic domain (MgtE 1–275). Both endogenous and ectopically expressed YqgP cleaved endogenous MgtE, yielding distinct cleavage products (Fig 2A). The cleavage was abrogated by a rhomboid-specific peptidyl ketoamide inhibitor (Ticha *et al*, 2017b) at low nanomolar levels, confirming that it was a rhomboid-specific event (Fig 2B). Judging by the apparent molecular size of the N-terminal cleavage fragment compared with the *in vitro*-translated reference fragments (Lemberg & Martoglio, 2003) of MgtE, we concluded that the cleavage by YqgP occurred within the extracytoplasmic loop between TMH1 and TMH2 of MgtE. This region is close to or within the periplasmic end of TMH2 (Fig 2C and D), which is consistent with the topology and mechanism of a rhomboid protease (Strisovsky, 2013, 2016). MgtE transporters function as homodimers with a cytosolic amino-terminal cystathionine-beta-synthase (CBS) domain that senses intracellular $Mg^{2+}$ and a carboxy-terminal five-transmembrane helical pore for $Mg^{2+}$ import (Hattori *et al*, 2007; Takeda *et al*, 2014). Cleavage of MgtE between TMH1 and TMH2 by YqgP would thus likely inactivate the transporter.

### Manganese excess activates cleavage of MgtE by YqgP under conditions of magnesium starvation

It is known that MgtE homologs can also transport $Mn^{2+}$ (Takeda *et al*, 2014) under low $Mg^{2+}$ concentration (Mg starvation) and that high concentrations of $Mn^{2+}$ are toxic for bacteria, mainly because $Mn^{2+}$ can mis-metalate $Mg^{2+}$-binding sites (Hohle & O'Brian, 2014; Chandrangsu *et al*, 2017). We therefore hypothesised that YqgP

could be involved in the regulation of MgtE under Mn²⁺ stress, and we tested the influence of manganese on the *in vivo* activity of YqgP. While at high extracellular magnesium concentration (1 mM) the effect of 100 μM MnCl₂ was not detectable, at low extracellular magnesium concentration (0.01 mM), when endogenous MgtE is upregulated, a shift from 1 to 100 μM MnCl₂ activated the YqgP-dependent cleavage of MgtE fourfold (Fig 3A).

We next tested the importance of YqgP for growth in manganese-stress conditions. During growth in minimal medium at low magnesium concentration (10 μM), adding 75 μM manganese at mid-log phase caused growth arrest and lysis of wild-type *B. subtilis*, which was more severe upon deletion of *yqgP*, and fully rescued by ectopic expression of YqgP (Fig 3B). This effect was highly reproducible. Interestingly, all strains resumed growth

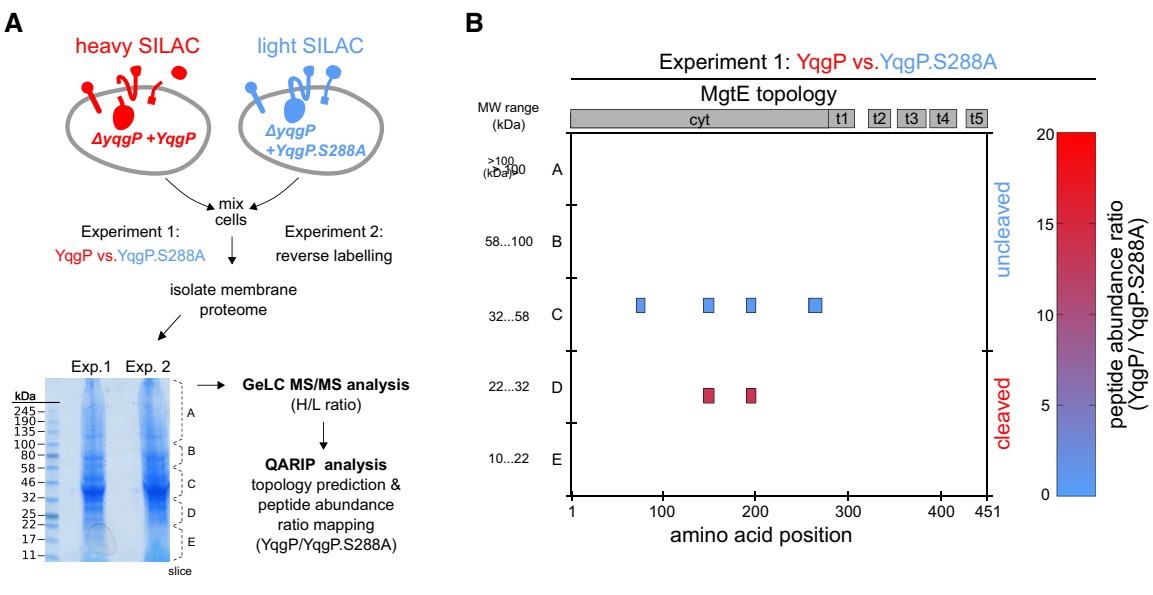

C

| Gel slice | MW range (kDa) | Uniprot accesion | Gene name | MW theor. (kDa) | Protein | Topology prediction (Phobius) | TM domains | Identified peptides | Abundance ratio for the intracellular part |
|---|---|---|---|---|---|---|---|---|---|
| Exp2_D | 22 to 32 | O34442 | mgtE | 50.8 | Magnesium transporter MgtE | Nin-Cout | 5 | 2 | 17.94 |
| Exp1_C | 32 to 58 | O31707 | yknU | 66.2 | Uncharacterized ABC transporter ATP-binding protein YknU | Nin-Cout | 6 | 1 | 17.49 |
| Exp1_D | 22 to 32 | O34442 | mgtE | 50.8 | Magnesium transporter MgtE | Nin-Cout | 5 | 2 | 13.45 |
| Exp2_C | 32 to 58 | O07549 | yheH | 76.3 | Probable ABC transporter ATP-binding protein YheH | Nout-Cout | 4 | 1 | 2.16 |
| Exp1_C | 32 to 58 | P20166 | ptsG | 75.5 | Glucose-specific phosphotransferase enzyme IIA component | Nout-Cin | 11 | 1 | 2.07 |
| Exp2_E | 10 to 22 | P54467 | yqfB | 15.7 | Uncharacterized protein YqfB | Nout-Cin | 1 | 1 | 2 |

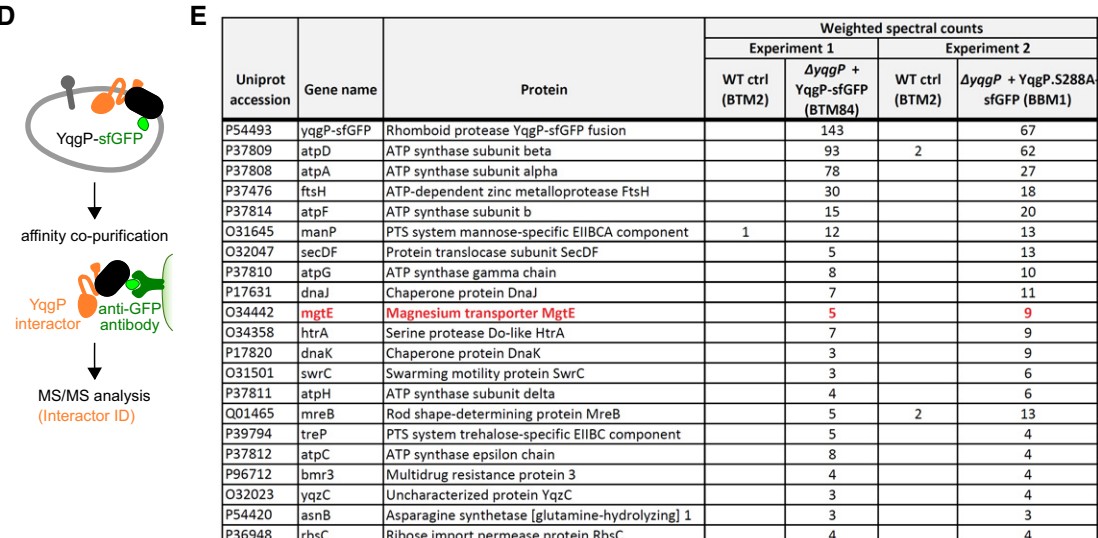

E

| Uniprot accession | Gene name | Protein | Weighted spectral counts | | | |
|---|---|---|---|---|---|---|
| | | | Experiment 1 | | Experiment 2 | |
| | | | WT ctrl (BTM2) | ΔyqgP + YqgP-sfGFP (BTM84) | WT ctrl (BTM2) | ΔyqgP + YqgP.S288A-sfGFP (BBM1) |
| P54493 | yqgP-sfGFP | Rhomboid protease YqgP-sfGFP fusion | | 143 | | 67 |
| P37809 | atpD | ATP synthase subunit beta | | 93 | 2 | 62 |
| P37808 | atpA | ATP synthase subunit alpha | | 78 | | 27 |
| P37476 | ftsH | ATP-dependent zinc metalloprotease FtsH | | 30 | | 18 |
| P37814 | atpF | ATP synthase subunit b | | 15 | | 20 |
| O31645 | manP | PTS system mannose-specific EIIBCA component | 1 | 12 | | 13 |
| O32047 | secDF | Protein translocase subunit SecDF | | 5 | | 13 |
| P37810 | atpG | ATP synthase gamma chain | | 8 | | 10 |
| P17631 | dnaJ | Chaperone protein DnaJ | | 7 | | 11 |
| O34442 | mgtE | Magnesium transporter MgtE | | 5 | | 9 |
| O34358 | htrA | Serine protease Do-like HtrA | | 7 | | 9 |
| P17820 | dnaK | Chaperone protein DnaK | | 3 | | 9 |
| O31501 | swrC | Swarming motility protein SwrC | | 3 | | 6 |
| P37811 | atpH | ATP synthase subunit delta | | 4 | | 6 |
| Q01465 | mreB | Rod shape-determining protein MreB | | 5 | 2 | 13 |
| P39794 | treP | PTS system trehalose-specific EIIBC component | | 5 | | 4 |
| P37812 | atpC | ATP synthase epsilon chain | | 8 | | 4 |
| P96712 | bmr3 | Multidrug resistance protein 3 | | 4 | | 4 |
| O32023 | yqzC | Uncharacterized protein YqzC | | 3 | | 4 |
| P54420 | asnB | Asparagine synthetase [glutamine-hydrolyzing] 1 | | 3 | | 3 |
| P36948 | rbsC | Ribose import permease protein RbsC | | 4 | | 4 |

**Figure 1.**

◄

**Figure 1. Quantitative proteomics reveals candidate interactors and substrates of rhomboid protease YqgP from *Bacillus subtilis*.**

A   Schematic representation of the SILAC-based quantitative proteomic experiment in *B. subtilis*. Cells overexpressing active YqgP (BS50, Table EV2) or its catalytically dead mutant YqgP.S288A (BS51), both auxotrophic for lysine, were grown in parallel in "heavy" (containing $^{13}C_6^{15}N_2$-lysine isotope) or "light" (containing stable $^{12}C_6^{14}N_2$-lysine) M9 minimal medium, respectively. After mixing the cell cultures in the 1:1 ratio (based on $OD_{600}$), cell suspension was lysed and the fraction enriched for transmembrane proteome was analysed in a GeLC-MS/MS experiment. Using bioinformatic analysis of the MS data, highest-scoring candidate substrates were further evaluated.

B   Diagrammatic representation of a result of GeLC-MS/MS analysis of excised gel regions described in (A). Each coloured block depicts a quantified peptide at its amino acid position according to the horizontal axis. Possible ongoing proteolysis was identified by high abundance ratio of YqgP/YqgP.S288A for a given protein and lower apparent molecular weight than expected for the corresponding full-length protein, as estimated from the position of the respective gel slice relative to the molecular weight marker.

C   Table of best substrate candidates of two GeLC experiments. In order to assess possible cleavage sites in combination with topology information, the QARIP software (Ivankov *et al*, 2013) was used to summarise results from Experiments 1 and 2. Abundance ratios (YqgP/YqgP.S288A) for the intracellular parts of proteins were calculated for each gel slice separately by QARIP from peptide ratios computed by MaxQuant (Cox & Mann, 2008). Substrate candidates highlighted in red were identified in both experiments.

D   Schematic representation of the affinity co-purification experiment in *B. subtilis* to identify YqgP interactors.

E   Results of MS analyses of affinity co-purification experiments in wild-type *B. subtilis* control (strain BTM2, Table EV2) and *B. subtilis* deficient in endogenous YqgP expressing the wild-type YqgP-sfGFP bait (strain BTM84, Table EV2) or the proteolytically inactive YqgP.S288A-sfGFP bait (strain BBM1, Table EV2). Proteins were considered as potential interactors of YqgP if they were identified only in both positive co-purifications with a minimum of three weighted spectral counts or enriched at least five times in positive bait samples compared with control ones based on weighted spectral counts. The protein highlighted in red was the only overlapping hit between the two proteomic approaches.

at later time points, indicating the possible existence of an adaptive mechanism. This did not appear to involve the second *B. subtilis* rhomboid YdcA, because deletion of *ydcA* did not modify the phenotypic behaviour of *B. subtilis* regardless of the presence or absence of *yqgP* (Fig EV1). Regardless, the *yqgP*-deficient strains showed a marked growth disadvantage even in the latter phase of the growth curve, indicating that the role of YqgP

for the fitness of *B. subtilis* in these conditions is significant. The toxic effect of 50 μM manganese sulphate was fully prevented by growth in high (5 mM) magnesium sulphate (Fig 3C), which was consistent with a similar effect on MgtE cleavage by YqgP (Fig 3A). In addition, increasing concentrations of YqgP inhibitor had a dose-dependent effect on the protective role of YqgP both when it was overexpressed or present at endogenous levels

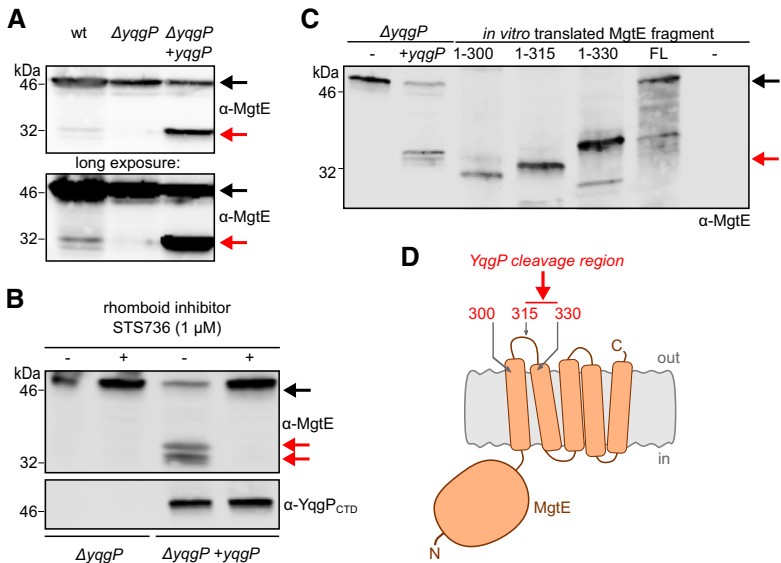

**Figure 2. YqgP cleaves the high-affinity magnesium transporter MgtE between its first and second transmembrane helices.**

A   Steady-state cleavage profile of endogenous MgtE processed by endogenous and ectopically overexpressed YqgP from the inducible $P_{hyperspank}$ promoter in living *Bacillus subtilis* (BTM2 and BTM501, respectively, Table EV2), in minimal medium at low magnesium concentration (10 μM). Strain lacking YqgP (ΔyqgP, BTM78, Table EV2) was used as a control. Proteins were detected by immunoblotting with chemiluminescence detection.

B   The cleavage is efficiently inhibited by 1 μM STS736, a specific peptidyl ketoamide rhomboid inhibitor (Ticha *et al*, 2017b).

C   To map the cleavage site region within endogenous MgtE (second lane from the left, from strain BTM501), MgtE-derived reference fragments encoding first 300, 315 and 330 amino acids, as well as full-length MgtE (black arrow), were *in vitro*-transcribed and *in vitro*-translated. Mobility of the N-terminal cleavage product (red arrow) of MgtE on SDS–PAGE was compared to the mobilities of the translated reference fragments.

D   Diagrammatic display of the mapping shows that YqgP cleaves MgtE in a periplasmic region between transmembrane helices 1 and 2 (red arrow).

Data information: In all panels, endogenous full-length MgtE is indicated by a black arrow, and N-terminal cleavage product by red arrows. Endogenous MgtE was visualised by anti-MgtE(2–275) (α-MgtE) and ectopic YqgP by anti-YqgP antibodies.

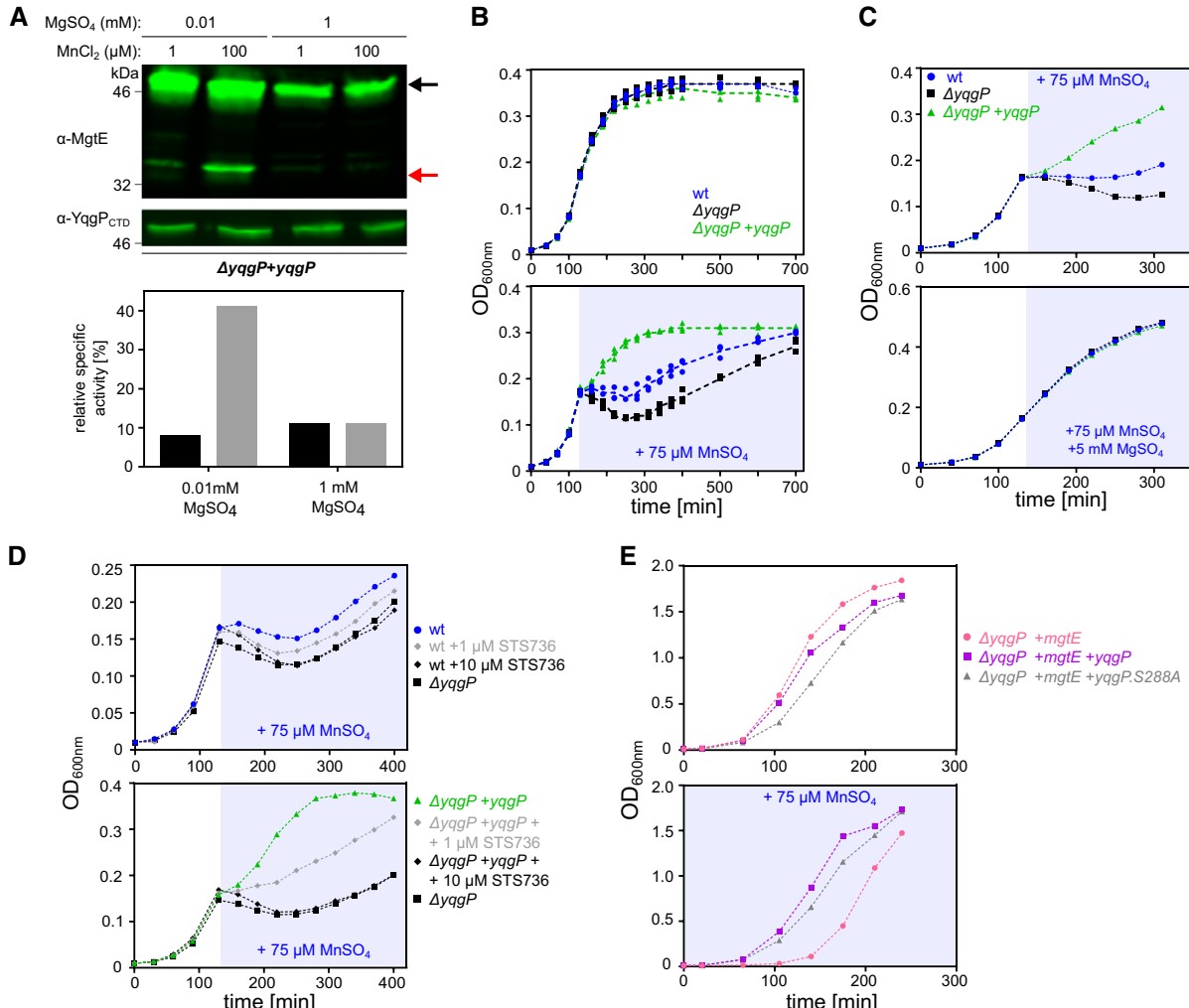

**Figure 3. Magnesium starvation and manganese excess activate cleavage of MgtE by YqgP, which is beneficial in manganese-stress conditions.**

A  Detection and quantification of the cleavage of endogenous MgtE by YqgP in living *Bacillus subtilis* cells (BS72, Table EV2) depending on the concentrations of magnesium and manganese ions. Cells were cultivated in glucose M9 minimal medium with limiting (0.01 mM) or high (1 mM) concentration of MgSO₄, in the presence or absence of 100 μM MnCl₂, and analysed by Western blotting with near-infrared detection (upper panel). Black arrow denotes full-length MgtE, and red arrow denotes its N-terminal cleavage product formed by YqgP. The corresponding fluorescence signals were quantified by densitometry, and are displayed as relative specific activity, which is substrate conversion normalised to enzyme expression level (lower panel).

B  Growth curves of wild-type (BTM843, Table EV2), *yqgP*-deficient (BTM844, Table EV2) and rescue (BTM845, Table EV2) strains of *B. subtilis* in M9 minimal medium with limiting magnesium (0.01 mM MgSO₄), exposed to manganese stress elicited by adding 75 μM MnSO₄ in mid-exponential phase (stress phase denoted by blueish background). All strains further contain a deletion in the putative manganese efflux pump MntP (*ΔywlD*, Table EV2). Bottom panel shows that manganese is more toxic in the *yqgP*-deficient strain than in the wild-type strain and that reintroduction of YqgP rescues fitness during manganese stress to above wild-type level. Top panel shows no difference between the strains in the absence of manganese stress. Data are shown as individual datapoints from three independent experiments overlaid with dashed line connecting average values from each time point, which illustrates the reproducibility of the assay.

C  Top panel: growth curves of wild-type (BTM843), *yqgP*-deficient (BTM844) and rescue (BTM845) strains of *B. subtilis* in M9 minimal medium with limiting magnesium (0.01 mM MgSO₄), exposed to manganese shock elicited by adding 75 μM MnSO₄ in mid-exponential phase. Bottom panel: manganese toxicity is prevented by further adding 5 mM magnesium (MgSO₄) in otherwise identical conditions.

D  Inhibition of YqgP by a rhomboid-specific peptidyl ketoamide inhibitor (STS736, i.e. compound **9** from Ticha et al, 2017b) abolishes the YqgP-induced fitness of *B. subtilis* under manganese stress, in a dose-dependent manner, both with endogenous YqgP (top panel) and overexpressed YqgP (bottom panel). Media and growth conditions were identical to those used in panel (B).

E  Overexpression of heterologous MgtE inhibits growth of *yqgP*-deficient strain (BTM610, Table EV2) in rich LB medium supplemented with 75 μM MnSO₄. Cell fitness is improved by overexpression of YqgP (BTM611, Table EV2) or its catalytically dead mutant YqgP.S288A (BTM612, Table EV2). For clarity, for panels (C–E), representative experiments of 2–3 independent biological replicates are shown.

(Fig 3D), indicating that the pronounced protective effect of ectopically expressed YqgP was a result of its overexpression. Finally, the phenotype manifested also during steady-state growth in rich

LB medium supplemented with 75 μM manganese sulphate, where *yqgP*-deficient *B. subtilis* ectopically expressing MgtE was delayed in growth compared with the same strain ectopically expressing

YqgP (Fig 3E). Intriguingly, ectopic expression of a proteolytically inactive S288A mutant of YqgP also rescued the phenotype of the parent strain, which was initially surprising and suggested that YqgP may have a role independent of its protease activity. Together, these results reveal that a physiological function of YqgP is the protection from manganese stress by contributing to the degradation of the main magnesium transporter MgtE in *B. subtilis* (Wakeman *et al*, 2014).

### The N-terminal extramembrane intracellular domain of YqgP mediates the manganese-induced activation of MgtE cleavage

To gain insights into the mechanism of MgtE cleavage by YqgP, we turned our attention to the two extramembrane domains of YqgP, the N-terminal cytosolic domain (NTD) and the C-terminal extracellular domain (CTD). While CTD consists of a predicted tandem array of three tetratricopeptide repeat (TPR) motifs (Zeytuni & Zarivach, 2012) and is thus likely to interact with protein or peptide ligands, NTD did not show significant sequence similarity to any protein of known structure, providing no indication of its molecular function. We first investigated the importance of NTD and CTD for YqgP activity. We created *B. subtilis* strains producing YqgP lacking NTD or CTD and measured the *in vivo* "specific activities" of these YqgP variants (i.e. substrate conversions at steady state normalised to enzyme expression level) in cleaving MgtE and the heterologous model substrate TatA (Ticha *et al*, 2017a). While deletion of CTD (Δ388–507, ΔCTD) seems to have led to about four- to sixfold increase in YqgP activity for both substrates, deletion of NTD (Δ1–178, ΔNTD) reduced YqgP activity for MgtE threefold, but only marginally so for the model TatA substrate (Fig 4A). In addition, deletion of NTD also compromised the activation effect of $Mn^{2+}$ on MgtE cleavage by YqgP, while the deletion of CTD had no effect (Fig 4B–D). Taken together, the existence of such a specific regulatory mechanism involving YqgP NTD further substantiated the idea that the manganese-regulated cleavage of MgtE by YqgP is physiologically significant.

### The N-terminal intracellular domain of YqgP binds divalent cations and mediates their sensing by YqgP

Since divalent cations often have related coordination properties, we wondered whether the manganese activation effect on YqgP transmitted via $YqgP_{NTD}$ may be a reflection of a more general divalent metal cation effect. We thus inspected a range of divalent metal cations for activation of MgtE cleavage by YqgP *in vivo* and found that high concentrations (100 μM) of $Mn^{2+}$, $Zn^{2+}$ and $Co^{2+}$ activated MgtE cleavage by YqgP significantly in minimal medium with limiting $Mg^{2+}$ conditions (10 μM), while $Ca^{2+}$ and $Ni^{2+}$ did not have appreciable effect (Fig 5A). Isothermal titration calorimetry (ITC) measurements showed that $Mn^{2+}$, $Zn^{2+}$, $Co^{2+}$ and $Ni^{2+}$ bound $YqgP_{NTD}$ with dissociation constants in the submillimolar to millimolar range and approximately 1:1 stoichiometry, while neither $Mg^{2+}$ nor $Ca^{2+}$ showed any calorimetric effect up to 1.5 mM concentrations tested (Fig 5B). Interestingly, the transition metals $Co^{2+}$, $Mn^{2+}$ and $Zn^{2+}$ can all be transported by MgtE, albeit with lower efficiency than $Mg^{2+}$. Overexpression of MgtE confers high sensitivity to $Zn^{2+}$ in *Bacillus firmus* OF4 (Smith *et al*, 1995), and

$Ni^{2+}$ cannot be transported but inhibits MgtE transport (Smith *et al*, 1995). Here, we observed that *yqgP* deficiency made *B. subtilis* more sensitive to $Mn^{2+}$ and $Zn^{2+}$ but not to $Co^{2+}$ or $Ni^{2+}$ stress at low $Mg^{2+}$, and consistently, re-expression of YqgP rescued this effect to above wild-type levels (Fig 5C). Collectively, this indicates that $YqgP_{NTD}$ may act as a sensor for a subset of metal cations in order to prevent cation toxicity due to MgtE permissiveness under low $Mg^{2+}$ conditions.

To understand the basis of the activation of YqgP by divalent cations, we undertook structural analysis of $YqgP_{NTD}$. Domains homologous to $YqgP_{NTD}$ occur only in YqgP orthologs among the soil-dwelling bacteria of the genus *Bacilli* (http://www.ebi.ac.uk/interpro/protein/P54493/similar-proteins), bear little sequence similarity to other proteins and show no hits at ProSite. Thus, to understand the structural basis of manganese-induced activation of YqgP, we first determined the 3D structure of NTD by solution NMR (Fig 6A, statistics in Table EV3). The overall structural ensemble of $YqgP_{NTD}$ (YqgP 1–177) consists of both α- and β-structural motifs, comprising six β-strand sequences at positions 21–24 (β1), 30–34 (β2), 43–48 (β3), 80–88 (β4), 104–106 (β5) and 109–117 (β6), and five α-helices at positions 3–16 (α1), 54–75 (α2), 96–100 (α3), 121–128 (α4) and 146–168 (α5). The central mixed β-sheet motif is packed in a twisted plane and surrounded by helices α1–4. Notably, negatively charged side chains are clustered into several solvent-exposed regions (Fig 6B). Structural similarity search using Dali server (Holm & Laakso, 2016) identified a number of structurally related proteins, with no conserved functional role but enriched in DNA/RNA polymerases and modifying enzymes (Table EV4). In particular, the β-stranded central part of $YqgP_{NTD}$ adopts a fold characteristic for the type II restriction endonucleases (InterPro and Pfam codes IPR018573 and PF09491, respectively) (Kachalova *et al*, 2008).

Since $Mn^{2+}$ but not $Zn^{2+}$ provided reliable data in ITC titrations, we focused on $Mn^{2+}$ in further structural analyses aiming to understand the structural aspect of divalent cation-mediated activation via $YqgP_{NTD}$. NMR titrations of $YqgP_{NTD}$ by the paramagnetic $Mn^{2+}$ induced peak intensity reductions and chemical shift perturbations in a number of residues along NTD. N-H chemical shifts could be caused by direct interaction with the metal ion, or by secondary structural effects, while peak intensity could be also diminished by the paramagnetic effect of $Mn^{2+}$. The most strongly affected regions were amino acids 27–30, 35–37, 48–63 and 89–98 of the 177 amino acid NTD (Fig 6C and D, full titration data are available in Dataset EV3). To identify potential residues involved in binding $Mn^{2+}$, we next focused on amino acids typically involved in chelating divalent cations (D, E, H) from each of the regions affected in NMR titrations. We analysed the effect of alanine substitutions in D29, D37, H49, D50, D52, D60 and E90 on the ability of these mutants to elicit Mn-induced activation of YqgP *in vivo* (Fig 6E). This analysis indicated that the hotspot in which mutations interfered with Mn-induced activation of YqgP was centred around D50, and D50A mutation abrogated Mn activation completely. Interestingly, several of the residues affected in NMR titrations clustered in one apex of NTD structure, with D50 being central in this region (Fig 6F), representing the likely Mn-binding region in NTD (Fig 6G). In summary, deletion of $YqgP_{NTD}$ or mutation of chelating residues from the identified $Mn^{2+}$-binding surface abrogated the $Mn^{2+}$-induced activation of MgtE

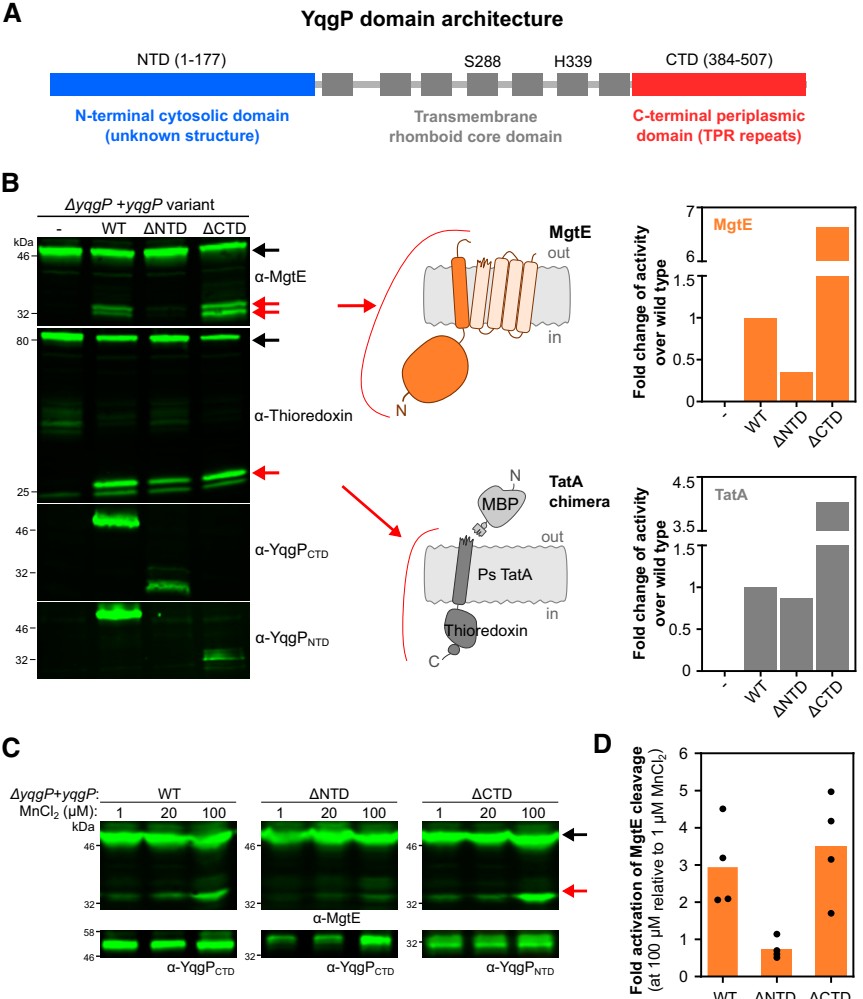

**Figure 4. The N-terminal extramembrane intracellular domain of YqgP mediates the manganese-induced activation of MgtE cleavage.**

A  Schematic of YqgP domain architecture. Transmembrane helices are shown as grey boxes, and the position of the catalytic dyad of S288 and H339 within the transmembrane rhomboid core is displayed.

B  Detection (left panel) and quantification (right panel) of steady-state conversion of endogenous MgtE and model chimeric substrate derived from *Providencia stuartii* TatA (Stevenson *et al*, 2007; Ticha *et al*, 2017a,b) (schemes in the middle) by ectopically expressed YqgP and its N- and C-terminally truncated variants (strains BS184–187, Table EV2) in living *Bacillus subtilis* grown in rich LB medium. Black arrows indicate full-length substrates, and red arrows indicate cleavage products.

C  Similar analysis of the same strains but grown in minimal M9 medium (with 10 μM MgSO₄) supplemented with increasing concentrations of MnCl₂. Black arrows denote full-length substrate (MgtE), and red arrows denote the cleavage products generated by YqgP.

D  Western blots of four independent experiments shown in panel (C) were quantified by near-infrared fluorescence detection, quantified by densitometry and the results are displayed as fold activation of MgtE cleavage at 100 μM MnCl₂ relative to 1 μM MnCl₂ activity. Average values and all fours datapoints are plotted for each indicated YqgP variant.

cleavage by YqgP *in vivo* showing that NTD is the $Mn^{2+}$-sensing device of YqgP.

## YqgP works in tandem with the AAA protease FtsH to degrade MgtE in response to the environmental divalent cation concentrations

To explore more precisely MgtE proteolysis *in vivo*, we decided to construct a *yqgP* mutant strain complemented with a ectopic mutated version of *yqgP* that encodes a catalytically dead YqgP.S288A protein. Surprisingly, we observed that the production of this inactive YqgP.S288A in the *yqgP*-deficient background

resulted in the production of an alternative band with lower mobility on SDS–PAGE, i.e. a higher molecular weight, than that of the cleavage product generated by YqgP (Fig 7A). Since the S288A variant of YqgP lacks the catalytic serine nucleophile and is thus proteolytically completely inactive (Lemberg *et al*, 2005; Wang *et al*, 2006; Baker & Urban, 2012), these results suggested that another proteolytic enzyme may participate in MgtE processing under these conditions. Co-immunoprecipitation experiments (Fig 1E) showed that YqgP interacts with the ATP-dependent, membrane-bound, processive zinc metalloprotease FtsH, and we thus hypothesised that FtsH is the responsible enzyme. Indeed, deletion of *ftsH* led to the disappearance of this alternative cleavage product (Fig 7A).

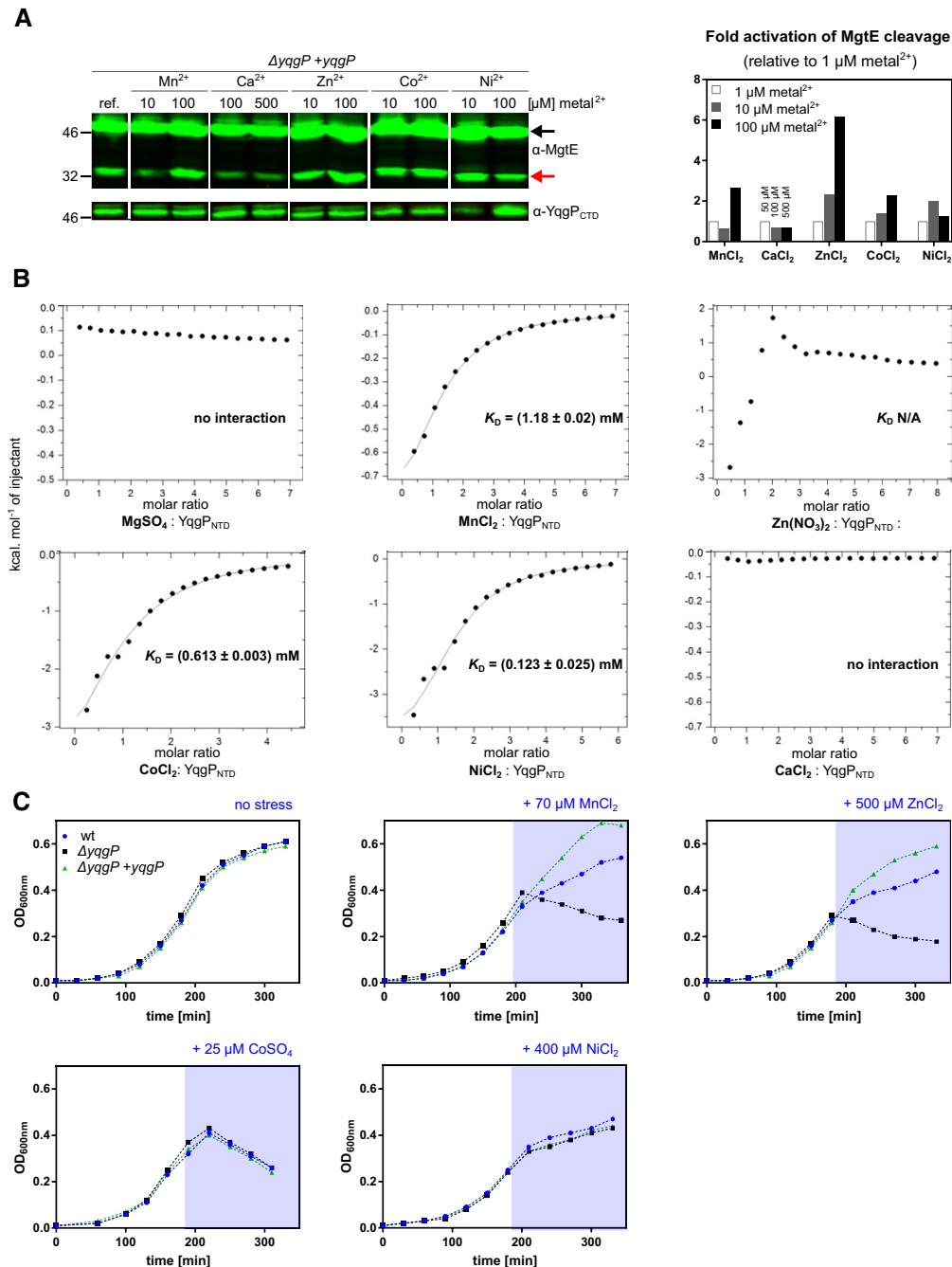

**Figure 5. The N-terminal intracellular globular domain of YqgP binds divalent cations.**

A  Detection (left panel) and quantification (right panel) of steady-state conversion of endogenous MgtE by YqgP (strain BS72, Table EV2) grown in minimal M9 medium containing 10 μM MgSO₄ and 1 μM each of MnCl₂, ZnCl₂, CoCl₂, NiCl₂ and CaCl₂ (reference conditions) with the additions of 10 μM or 100 μM of a given divalent cation salt solution (MnCl₂, ZnCl₂, CoCl₂ or NiCl₂), or 50 μM and 500 μM for CaCl₂. Western blots were quantified by near-infrared fluorescence detection (left), quantified by densitometry and displayed as relative specific activity (graph on the right), which is substrate conversion normalised to enzyme expression level. Black arrow denotes full-length substrate (MgtE), and red arrow denotes the N-terminal cleavage product(s) generated by YqgP. All bands originate from the same Western blot and identical treatment series. Irrelevant lanes have been cropped out for clarity, and the source blot is available online as Source Data.

B  Isothermal titration calorimetry of purified recombinant YqgP$_{NTD}$ and selected divalent cations.

C  Cation toxicity assays and their relationship to YqgP activity. Wild-type *Bacillus subtilis* (BTM2, Table EV2), its variant lacking YqgP (*ΔyqgP*, BTM78, Table EV2) and the rescue strain ectopically expressing YqgP (BTM501, Table EV2) were cultivated in minimal M9 medium containing 10 μM MgSO₄ and 1 μM each of MnCl₂, ZnCl₂, CoCl₂, NiCl₂ and CaCl₂ (i.e. reference conditions), and in mid-exponential phase were stressed by the addition of either 70 μM MnCl₂, 500 μM ZnCl₂, 25 μM CoSO₄ or 400 μM NiCl₂ (pale blue area). YqgP activity specifically improved cell fitness during manganese and zinc stress, while it had no effect on growth of cells cultivated in the presence of high cobalt and nickel concentrations. Representative experiments of 2–3 independent replicates are shown.

Source data are available online for this figure.

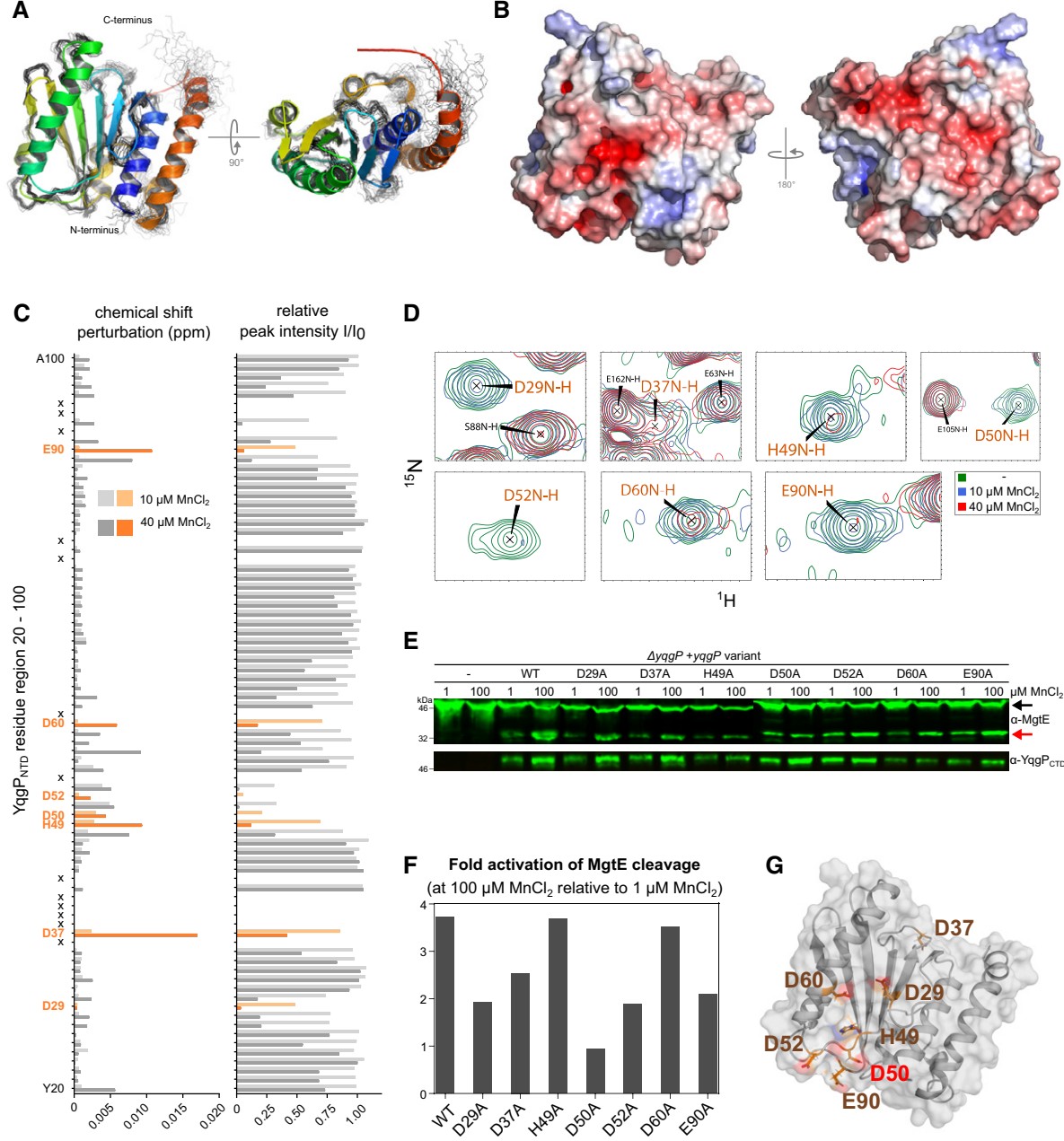

**Figure 6. Solution NMR analysis reveals manganese-binding region in the N-terminal domain of YqgP.**

A   Solution NMR structure of cytosolic N-terminal domain of YqgP (YqgP_NTD, amino acids 1–177), showing ensemble of 30 structures with lowest energy, and the structure with minimum energy displayed as cartoon.

B   Electrostatic surface of YqgP_NTD structure reveals highly negatively charged areas. Left view is oriented as the left view in panel (A).

C   Chemical shift perturbation values and relative peak intensities for backbone resonances ($^{15}$N and $^{1}$H) of YqgP_NTD calculated for complexes of 400 μM YqgP_NTD with 10 μM (light grey, light orange) or 40 μM MnCl_2 (dark grey, dark orange bars). Only residues 20–100 of YqgP are displayed for simplicity. Residues identified as potential manganese-binding region based on their chemical shifts and intensity changes upon NMR titrations are depicted in orange. The X sign marks residues whose chemical shifts were not calculated due to the lack of resonance signal.

D   Details of 2D $^{15}$N/$^{1}$H HSQC spectra of free (green) and Mn-titrated YqgP_NTD (10 μM MnCl_2 in red and 40 μM MnCl_2 in blue) for the residues marked in orange in panel (C). As a reference, residues marked in black do not display spectral changes upon titration, e.g. peak intensity does not change for S88N-H upon Mn titration (overlay of green, blue and red contours), but peak intensity is severely depleted for D29N-H upon Mn titration (absence of red contours). Full titration data are available in Dataset EV3.

E, F   Detection (E) and quantification (F) of steady-state conversions of endogenous MgtE by YqgP variants bearing single-point mutations in putative Mn-binding region (BS184; 187; 196–203, Table EV2), cultivated in modified M9 minimal medium supplemented by low (1 μM) or high (100 μM) MnCl_2. Black arrow marks full-length MgtE, and red arrow marks its N-terminal cleavage product by YqgP.

G   Overview of residues affected by Mn$^{2+}$ binding to YqgP_NTD mapped onto its solution NMR structure. The view is oriented as the left view in panels (A and B).

Experiment with reference fragments of MgtE indicated that the products of proteolysis by FtsH end between residues 340 and 355, which correspond to the cytosolic side of TMH 3 of MgtE (Fig 7B and C). This is consistent with the known mechanism of FtsH, which dislocates transmembrane proteins from the membrane and processively proteolyses them (Chiba *et al*, 2002; Akiyama, 2003; Langklotz *et al*, 2012; Yang *et al*, 2018). Interestingly, YqgP$_{NTD}$ is crucial also for this adaptor function of YqgP (Fig 7D).

To investigate the relationship between YqgP and FtsH in more detail, we performed chase experiments in which translation was inhibited by tetracycline at time-point zero ("translation shut-off"), which enabled us to observe the stability of the MgtE cleavage products generated by YqgP or FtsH (Fig 7E). This analysis revealed that during growth with ongoing translation, the cleavage products formed by YqgP (lanes 9–12 and lanes 33–36) or FtsH (lanes 17–20) accumulated. In contrast, no cleavage products accumulated after stopping translation. Instead, the YqgP-induced N-terminal cleavage product of MgtE was further degraded by FtsH (compare lanes 13–16 to lanes 37–40) but also by other, unknown proteases (lanes 37–40) (for quantification, see Fig EV2). In contrast, the alternative N-terminal cleavage product of MgtE generated by FtsH in the presence of YqgP.S288A was stable and was not further degraded over the chase period (lanes 21–24). In the absence of YqgP, FtsH was not able to act on MgtE at all (lanes 1–8). Intriguingly, this was mimicked by the expression of active YqgP in the presence of saturating concentrations of its specific peptidyl ketoamide inhibitor (compare Figs 2B and 7), which binds into the active site of rhomboid and plugs it (Ticha *et al*, 2017b). The above data mean that YqgP fulfils dual role, both of a protease and a substrate adaptor for FtsH, and that the formation of MgtE species susceptible to proteolysis by either protease requires ongoing translation. The adaptor function does not require YqgP activity, but it does require an unobstructed active site, and is thus consistent with a "pseudoprotease" function of YqgP. NTD is essential also for the pseudoprotease role of YqgP (Fig 7D), and without YqgP protease activity, FtsH cannot degrade MgtE fully (Fig 7), indicating that YqgP and FtsH closely co-operate in maintaining the proteostasis of MgtE.

In conclusion, the above data yield a model where YqgP responds to divalent metal cation concentrations and cleaves MgtE under low Mg and high Mn$^{2+}$ or Zn$^{2+}$ concentrations, facilitating the dislocation of MgtE cleavage products from the membrane and their degradation by FtsH (Fig 8). Hence, YqgP serves as a fail-safe mechanism to limit toxicity of metal cations quickly and temporarily. It has a dual role here, because besides cleaving its substrate MgtE, it also presents the cleavage products of MgtE to FtsH, which can then degrade them processively. Interestingly, while the proteolytically inactive YqgP.S288A variant allows formation of an alternative cleavage product by FtsH (i.e. it can still present MgtE to FtsH), this band is not observed when wild-type YqgP is expressed in the presence of a rhomboid protease inhibitor (Fig 2), suggesting that the active site of YqgP needs to be unobstructed to present MgtE (fragments) to FtsH. This phenomenon is remarkably similar to the roles of the rhomboid-like proteins Derlins, the p97 AAA ATPase and proteasome in ER-associated degradation (ERAD) (Ticha *et al*, 2018), and may constitute a prokaryotic precursor of ERAD. Furthermore, it provides a plausible mechanistic explanation of how the rhomboid-like pseudoproteases might have evolved from rhomboid proteases (Adrain & Freeman, 2012). In summary, we provide evidence for the role of a bacterial rhomboid protease in the regulation of key membrane protein proteostasis under environmentally restricted conditions in a process similar to the eukaryotic ERAD.

## Discussion

Metal ions are essential for life because they serve as cofactors in various cellular processes, but their excess poses a lethal threat. Hence, various mechanisms exist to ensure metal ion homeostasis (Chandrangsu *et al*, 2017). Soil bacteria, such as *B. subtilis*, frequently encounter fluctuations in metal ion availability and need to possess sophisticated regulatory and feedback mechanisms to specifically respond to a limitation or excess of various metal ions (Chandrangsu *et al*, 2017). Magnesium (Mg$^{2+}$) represents the most abundant divalent cation in cells. It is maintained at high free intracellular levels (0.5–1 mM) relative to other cations (Froschauer *et al*, 2004), and it plays vital roles in membrane integrity, charge neutralisation of nucleic acids and nucleotides, ribosome structure and activities of many metabolic enzymes. To take up environmental Mg$^{2+}$, bacteria usually encode several magnesium transporters of the CorA, MgtA and MgtE families (Groisman *et al*, 2013). Homologs of MgtE are universally conserved up to humans (SLC41 family), indicating their importance. In *B. subtilis*, MgtE is the main magnesium transporter (Wakeman *et al*, 2014) and is tightly regulated at several levels (Groisman *et al*, 2013). First, low levels of environmental magnesium trigger upregulation of *mgtE* transcription via a riboswitch mechanism (Dann *et al*, 2007). Second, the Mg-dependent gating mechanism by the N-terminal cytosolic domain of MgtE ensures metal ion selectivity (Hattori *et al*, 2009). Third, ATP further regulates MgtE (Tomita *et al*, 2017), presumably because the cell needs to maintain a balance in magnesium and ATP concentrations as the two are functionally coupled in a number of physiological processes (Pontes *et al*, 2015, 2016).

Structural principles of ion channel selectivity of MgtE have been studied in detail (Takeda *et al*, 2014; Maruyama *et al*, 2018). At low magnesium concentrations (when Mg$^{2+}$ is not a competitive inhibitor), MgtE can also transport Co$^{2+}$, Mn$^{2+}$ and Zn$^{2+}$ to some extent, while Ni$^{2+}$ inhibits MgtE (Smith *et al*, 1995; Takeda *et al*, 2014). This implies that at low magnesium concentrations and high concentrations of Co$^{2+}$, Mn$^{2+}$, Zn$^{2+}$ or Ni$^{2+}$, other regulatory mechanisms may be required to maintain the biological function of Mg and/or minimise toxicity of other divalent cations. Indeed, previous observations in other organisms indicate that controlling magnesium transporter selectivity with respect to transition metal ions is a widespread theme. In yeast, excess magnesium prevents manganese toxicity (Blackwell *et al*, 1997), implying that MgtE may transport manganese when magnesium does not compete with it. In *Bradyrhizobium japonicum*, a null mutant in the manganese transporter *mntH* encoding gene does not grow in Mn-limiting medium, but suppressor strains can be selected. One class of suppressors are gain-of-function mutants of *mgtE* (mutations affecting gating mechanism) (Hohle & O'Brian, 2014). In this system, when the intracellular Mg$^{2+}$ levels are low, the MgtE mutants transport Mn$^{2+}$ with high affinity, which is not the case at high Mg$^{2+}$ and for the wild-type strain. This is particularly relevant in soil, the natural habitat of *B. subtilis*, where there is a negative correlation between the bioavailability of magnesium and manganese ions. As soil pH

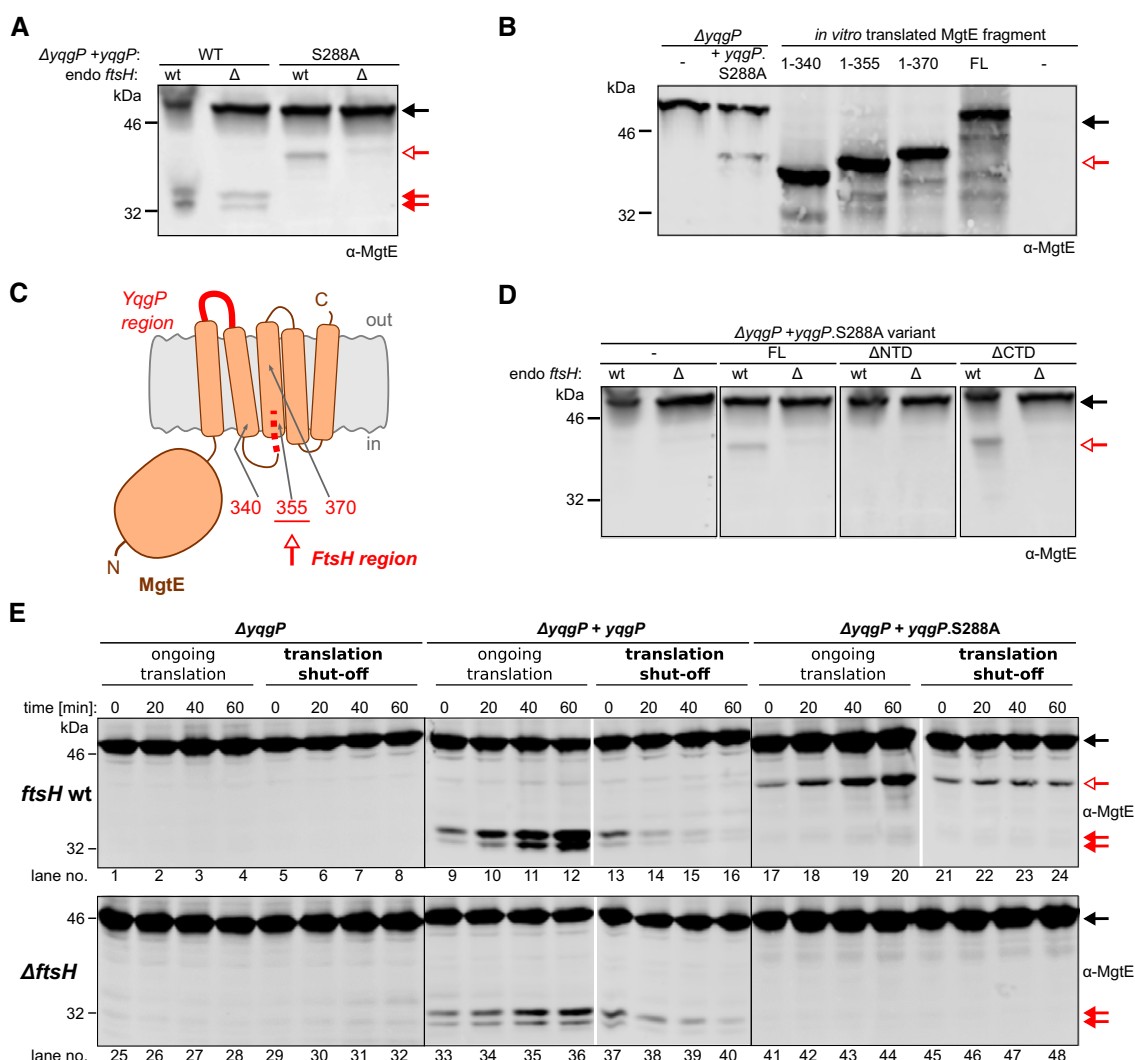

**Figure 7. The YqgP rhomboid functions as both a protease and a pseudoprotease adaptor of the AAA protease FtsH to maintain proteostasis of MgtE.**

A  The proteolytically dead mutant YqgP.S288A behaves as a pseudoprotease and acts as a substrate adaptor of FtsH. Pattern of C-terminally truncated forms of endogenous MgtE formed *in vivo* in the presence of ectopically expressed active YqgP (strain BS72, Table EV2) or its inactive S288A mutant (BS73, Table EV2) in rich LB medium in the presence or absence of endogenous FtsH. Black arrow denotes the full-length MgtE substrate, and red arrows denote its cleavage products generated by YqgP. While the presence of the faster migrating YqgP-dependent products (full red arrows) do not depend on the presence of FtsH, the slower migrating product formed in the presence of YqgP.S288A (open red arrow) does, indicating that FtsH protease is responsible for the alternative cleavage of MgtE. The same arrow symbolics are valid throughout this figure.

B  Mapping of the boundary of the FtsH generated proteolytic product of MgtE. Immunoblot comparing the SDS–PAGE mobility of the N-terminal fragment of endogenous MgtE generated by FtsH in the presence of the pseudoprotease version of YqgP (catalytically dead YqgP.S288A, strains BTM78, BTM502, Table EV2) to the mobilities of *in vitro*-translated reference fragments corresponding to MgtE 1–340, 1–355 and 1–370 shows that YqgP.S288A acts as an FtsH adaptor and that FtsH processing of MgtE in the presence of YqgP.S288A stops at around amino acid 355, which is near or at the cytoplasmic end of TMH 3 of MgtE.

C  Schematic depiction of the regions in MgtE defining its remaining cleavage products formed by FtsH and YqgP.

D  *In vivo* cleavage of endogenous MgtE in the presence of ectopically expressed full-length YqgP.S288A (strain BS73), YqgPΔNTD.S288A (BS185, Table EV2) or YqgPΔCTD.S288A (BS186, Table EV2), in rich LB medium shows that YqgP$_{NTD}$ is necessary for the FtsH-dependent processing of MgtE.

E  Analysis of the *in vivo* kinetics of formation and fate of MgtE cleavage products. *Bacillus subtilis* strains expressing YqgP or its S288A mutant (strains BTM78; 501 and 502, Table EV2) in the presence or absence of endogenous FtsH protease (strains BTM795, BTM796 and BTM797, Table EV2) were cultured in LB medium and at early exponential phase, after having been expressing YqgP variants for 30 min, were either left grown untreated (as a control with ongoing translation) or treated with 20 μg/ml tetracycline to stop proteosynthesis (translation shut-off). At given time intervals afterwards, all cultures were analysed by α-MgtE Western blotting with near-infrared detection. Equal cell number (judged by OD$_{600}$) was loaded into each lane. Black arrow marks full-length MgtE, red full arrows mark YqgP-dependent and red open arrows mark YqgP.S288A-dependent cleavage product of MgtE.

decreases, manganese becomes more available and magnesium less available (Thomas & Sprenger, 2008; Gransee & Fuhrs, 2013). Hence, in the commonly found acidic soils, *B. subtilis* has to cope

with stress conditions of low magnesium and high manganese, and the regulation of MgtE by YqgP that we describe is important for fitness in similar conditions (Figs 3 and 5).

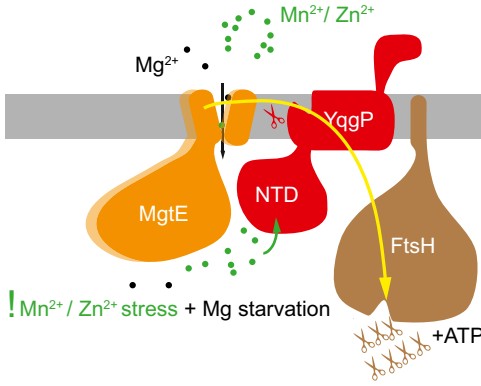

**Figure 8.  Summary of the role of YqgP in MgtE proteostasis.**

Under conditions of magnesium starvation, MgtE is upregulated and open, poised to transport $Mg^{2+}$ inside the cell. If other divalent cations, such as $Mn^{2+}$ or $Zn^{2+}$, are present in the environment at relatively high concentrations, MgtE can transport them or be inhibited by them, which causes toxicity. Under these conditions, the N-terminal cytosolic domain of YqgP binds $Mn^{2+}$ or $Zn^{2+}$, and activates YqgP for cleavage of MgtE between its transmembrane helices 1 and 2. YqgP also interacts with FtsH, and presents MgtE or its cleavage product(s) for processive proteolysis by FtsH. The cleavage of MgtE by YqgP is required for its full dislocation and degradation by FtsH. The presentation/adaptor function of YqgP is not dependent on its catalytic activity, but it requires its active site unoccupied. YqgP thus behaves also as a pseudoprotease which is common in the rhomboid superfamily. The connection of a rhomboid-like protease/ pseudoprotease (YqgP) with a processive degradative membrane-bound protease/dislocase (FtsH) and its role in membrane protein quality control in *Bacillus subtilis* that we identify here represent a striking analogy to the eukaryotic role of rhomboid-like pseudoproteases Derlins in ER-associated degradation.

The role of YqgP seems to be important during acute changes in cation concentrations. Several hours after manganese shock, even *yqgP*-deficient *B. subtilis* starts growing again, albeit at lower rate than wild type or a strain overexpressing YqgP (Fig 3B). This "adaptation" may be based on the described genetic mechanisms related to the regulation of Mn metabolism. It is known that *B. subtilis* responds to $Mn^{2+}$ excess in two ways. First, the MntR regulator senses excess intracellular $Mn^{2+}$. It then represses the transcription of two $Mn^{2+}$-uptake systems (MntABCD and MntH) and activates the expression of efflux systems (MneP and MneS) (Que & Helmann, 2000; Huang *et al*, 2017). In addition, the two manganese efflux pumps, MneP and MneS, are also activated via $Mn^{2+}$-responsive riboswitches (Dambach *et al*, 2015; Price *et al*, 2015). These actions take place at the transcriptional level, hence take some time to efficiently allow the cell to recover from an initial $Mn^{2+}$ shock. As intracellular $Mn^{2+}$ returns to physiological levels, cells resume growth and YqgP may thus lose its importance for fitness in the later stages of adaptation to these toxic conditions.

We find that in *B. subtilis*, rhomboid protease YqgP is required to cleave and inactivate species of MgtE formed at low environmental $Mg^{2+}$ and high $Mn^{2+}$ or $Zn^{2+}$, which protects *B. subtilis* from toxic $Mn^{2+}$ or $Zn^{2+}$ stress. The identity of the MgtE species susceptible to YqgP cleavage in these conditions is unknown. Since their generation requires ongoing translation (Fig 7), we hypothesise that these may be mis-metallated forms at some of the metal-binding sites throughout the MgtE molecule (Foster *et al*, 2014; Takeda *et al*, 2014; Maruyama *et al*, 2018) that can form during translation

and folding, and which could interfere with the proper channel function of MgtE. YqgP is thus providing a quality control check for properly functioning MgtE. Cleavage of MgtE by YqgP is more pronounced at low $Mg^{2+}$ and high $Mn^{2+}$ or $Zn^{2+}$ concentrations. This may, in part, result from the higher concentration of susceptible forms of MgtE in these conditions, which is consistent with our translation shut-off chase data (Fig 7). However, metal sensing and cleavage activation are also directly provided by the intracellular N-terminal extramembrane domain of YqgP. This domain binds divalent metal cations at millimolar or submillimolar affinities and approximately 1:1 stoichiometry (Fig 5) along a negatively charged surface of the molecule (Fig 6). How exactly in structural and mechanistic terms $Mn^{2+}$ or $Zn^{2+}$ binding to $YqgP_{NTD}$ activates MgtE cleavage is not clear at the moment. It could be through direct interaction with the membrane-embedded MgtE, or by influencing the speed of diffusion of YqgP in the membrane, as recently suggested for several other rhomboid proteases (Kreutzberger *et al*, 2019). Interestingly, in addition to the influence on YqgP activity, $YqgP_{NTD}$ is also required for the coupling of YqgP activity to FtsH, the ATP-dependent processive protease that associates with YqgP. More generally, there is emerging evidence from this work and the accompanying back-to-back study by Liu *et al* that rhomboid proteases can participate in membrane protein quality control in bacteria, where they are very widespread. It remains to be elucidated which structural and dynamic aspects of transmembrane domains rhomboid proteases can recognise in their substrates, to serve as "detectors" of misfolded or orphan protein species and adaptors of downstream degradative proteases. The variability in substrate specificities and various extramembrane domains of rhomboids may be the basis of their diverse functions in bacteria.

The specialised role that YqgP plays in the quality control of an essential transporter protein may seem unexpected, but conceptually, transporters are indeed regulated at a number of levels and very finely using specialised mechanisms. For example, amino acid permeases of the major facilitator superfamily in yeast are downregulated from the cell surface under high nutrient conditions by an increase in ubiquitin E3 ligase activity of Rsp5, which initiates their trafficking from the plasma membrane to multi-vesicular bodies (Lauwers *et al*, 2010). Other member of the same transporter family in yeast requires a specialised membrane chaperone Shr3, which assists the folding of the transporters and prevents their efficient degradation by ERAD (Kota *et al*, 2007).

Intriguingly, in addition to the role of YqgP in regulating MgtE, we suspect it can be engaged in other functions. In co-immunoprecipitation experiments with YqgP, we identified a number of ATP synthase subunits with high confidence and abundance by mass spectrometry (Fig 1E). Since we have not identified any of these subunits in our substrate screen (Fig 1C), we did not follow them up. However, this finding raises the possibility that further to its role in MgtE proteostasis, YqgP may be involved in the relationship between magnesium homeostasis and ATP synthesis, which would be consistent with a growing body of evidence supporting a regulatory relationship between magnesium and ATP levels. First, it was reported that a magnesium transporter-defective strain of *B. subtilis* can be rescued by inactivation of ATP synthase-encoding genes (Wakeman *et al*, 2014), suggesting that lowering ATP levels allows the cell to cope with magnesium deprivation. This is consistent with findings from Groisman and colleagues in *Salmonella enterica*,

where low levels of $Mg^{2+}$ in the cell lead to ATP synthase inhibition and reduced ribosome biogenesis. As a result, the pool of magnesium that is not ATP-bound anymore can be employed to support low levels of ribosome assembly, a priority process that requires large amounts of magnesium (Pontes *et al*, 2016). Finally, in the same bacterium, ATP synthase activity can be directly inhibited by the MgtC protein, whose expression is upregulated during magnesium deprivation (Lee *et al*, 2013). Strikingly, MgtC downregulation in the membrane of *S. enterica* is somewhat reminiscent of how MgtE is controlled by YqgP and FtsH in *B. subtilis*. MgtR, a small hydrophobic peptide, is thought to destabilise MgtC prior to degradation by FtsH (Alix & Blanc-Potard, 2008). Hence, during magnesium deprivation and cation toxicity in *B. subtilis*, it is tempting to speculate that YqgP may also inhibit the activity of ATP synthase to preserve the free $Mg^{2+}$ pool for translation purposes while assisting FtsH for the degradation of MgtE.

A number of native and model rhomboid substrates are single-pass membrane proteins (Urban *et al*, 2002a; Strisovsky *et al*, 2009; Riestra *et al*, 2015; Johnson *et al*, 2017; Saita *et al*, 2017). The cleavage of the transmembrane core of MgtE by YqgP described here adds to growing evidence that rhomboid proteases do engage in cleaving polytopic membrane proteins under some conditions (Erez & Bibi, 2009), although the features of transmembrane domains that they recognise still need to be properly structurally understood. Our results show that degradation of the susceptible MgtE species requires both YqgP and FtsH, with YqgP cleavage of the transmembrane core of MgtE being required for full degradation by FtsH. In addition, YqgP presents MgtE to FtsH, for which YqgP activity is dispensable, but its unoccupied active site is essential, thus behaving partially also as a pseudoprotease. In summary, here we uncover an ancestral membrane-associated degradation mechanism including a substrate receptor/protease and processive degradative dislocase and protease. This is conceptually remarkably similar to the process of ERAD in which rhomboid pseudoproteases Derlins co-operate with ubiquitin E3 ligases such as Hrd1, the AAA ATPase p97/VCP and the proteasome, in dislocating transmembrane proteins from the membranes of endoplasmic reticulum and degrading them (Neal *et al*, 2018; Ticha *et al*, 2018). Our work emphasises that even the active rhomboid proteases can have pseudoprotease functions, and suggests that the balance between their protease and pseudoprotease roles may vary. In this context, it is interesting to note that the second rhomboid protease encoded in *B. subtilis* genome, YdcA, lacks any detectable protease activity despite having all sequence hallmarks of an active rhomboid protease (Urban *et al*, 2002b; Lemberg *et al*, 2005). Similarly, mammalian RHBDL1 and RHBDL3 failed to cleave any known model substrates of rhomboid proteases (Johnson *et al*, 2017), and it ought to be borne in mind that in these or other cases, the pseudoprotease roles may be key to the physiological functions of these rhomboid proteins. More generally, other intramembrane proteases may have physiological roles that do not depend on their catalytic activity, such as the role of SPP in membrane protein dislocation from the ER (Loureiro *et al*, 2006).

A number of bacteria encode YqgP homologs (http://www.ebi.ac.uk/interpro/protein/P54493/similar-proteins), and the function of YqgP in magnesium homeostasis control that we describe here might be conserved more widely, because MgtE homologs are also very widespread (https://www.ebi.ac.uk/interpro/protein/O34442/

similar-proteins), although it is possible that YqgP homologs will have also other functions. This is indeed suggested by two recent reports. First, it was shown that the *Staphylococcus aureus* YqgP homolog Rbd is synthetic lethal with the nucleoid occlusion protein Noc and is involved in the initiation of DNA replication (Pang *et al*, 2017). Second, in a recent archived report, the same protein (although termed ActH therein) has been identified as a regulatory component of a complex with peptidoglycan amidase LytH and is required for LytH activation (Do *et al*, 2020). In neither case was it established whether the described function of *S. aureus* YqgP homolog Rbd/ActH depends on its protease activity. It is intriguing to speculate whether these functions will rely on a similar protease or pseudoprotease roles within a degradation machinery such as the one we describe here. It is fully conceivable that YqgP homologs can fulfil a range of functions in diverse bacteria depending on the client protein but using analogous general mechanism as we outline here.

In a generalised summary, this work implicates bacterial rhomboid proteases in the quality control of polytopic membrane proteins in cooperation with other processive proteases, which could be regarded as an evolutionary functional ancestor of the eukaryotic ER-associated degradation (ERAD). In our system, rhomboid plays not only the role of a protease that facilitates dislocation of the membrane protein by the ATP-dependent AAA protease FtsH, but also the role of a substrate adaptor handing over the substrates to FtsH. It is conceivable that with a more efficient and specialised dislocation machinery in higher organisms, the requirement for protease activity of the rhomboid-like protein (Derlin) could become less important, while the adaptor function becoming key. This logic would explain the particularly high occurrence of pseudoproteases in the rhomboid superfamily, which includes Derlins that are well-known essential components of the eukaryotic substrate recognition and dislocation machinery in ERAD (Neal *et al*, 2018).

# Materials and Methods

### Materials and chemicals

All common chemicals were from Sigma-Aldrich or VWR unless otherwise indicated. Oligonucleotides were from Sigma-Aldrich and Eurogentec; and restriction and DNA modification enzymes were from New England Biolabs.

### Plasmids and DNA cloning

All DNA constructs were created by PCR and isothermal assembly (Gibson, 2011), and verified by Sanger sequencing. All constructs used in this work are summarised in Table EV1.

### Bacterial strains, media and growth conditions

All *B. subtilis* strains used in this study were derived from the *168 trpC2+* strain (Nicolas *et al*, 2012; Koo *et al*, 2017), and construction details of the strains are listed in Table EV2. For routine work, *B. subtilis* strains were streaked onto solid LB agar plate (supplemented with appropriate antibiotic) from −80°C stock solution and

grown overnight at 37°C. Subsequently, single colony was inoculated into the fresh LB medium and cultivated until the pre-culture reached $OD_{600}$ around 1.0. Culture was then diluted in the fresh growth medium (LB or M9) to starting $OD_{600}$ of 0.01–0.025 and cultivated until the exponential phase was reached at $OD_{600}$ ranging from 0.6 to 1.5. Depending on the experimental setup, protein expression from an ectopic site was induced either by adding 1% (w/v) D-xylose (for constructs under the control of xylose promoter, $P_{xyl}$) or IPTG ($P_{hyperspank}$ promoter) at the start of cultivation as indicated, at 37°C, until $OD_{600}$ reached 1, unless stated otherwise. For regular growth, both *B. subtilis* and *E. coli* cells were cultivated in LB medium (Merck) at 37°C. For *in vivo* activity assays, *B. subtilis* cells were cultivated in modified M9 minimal medium (low or high $Mg^{2+}$ content) composed of M9 salt solution (6.8 g/l $Na_2HPO_4.2H_2O$, 3 g/l $KH_2PO_4$, pH 7.4, 0.5 g/l NaCl, 1 g/l $NH_4Cl$), 0.5% (w/v) D-glucose, 10 μM (low $Mg^{2+}$) or 1 mM $MgSO_4$ (high $Mg^{2+}$), 1 μM $MnCl_2$, 100 μM $CaCl_2$, 10 μM $FeSO_4$ and 18 amino acids including all genetically encoded ones except lysine and tyrosine (Harwood & Cutting, 1990). All minimal media in which toxicity assays were performed were prepared in water of ultrapure quality, processed through Milli-Q® Reference Water Purification System (Merck) and having resistivity of at least 18.2 MΩ.cm (at 25°C) and TOC values below 5 ppb. For translational shut-off chase experiments, *B. subtilis* cultures were grown to mid-exponential phase, in LB medium at 30°C. At this point, expression of *yqgP* from the $P_{hyperspank}$ promoter was induced by adding 0.1 mM IPTG for 30 min. At a given time point, proteosynthesis was blocked by the addition of tetracycline to a final concentration of 20 μg/ml. Cation toxicity assays were performed by adding concentrations of $MnCl_2$, $ZnCl_2$, $CoSO_4$ or $NiCl_2$ into M9 minimal medium at various growth time points, as specified in the result section. *Bacilli* cultures for SILAC-based proteomic experiments were cultured in modified M9 minimal medium (see above) supplemented with either naturally occurring "light" ($^{12}C_6{}^{14}N_2$) or "heavy" ($^{13}C_6{}^{15}N_2$) isotopic variants of L-lysine (Silantes GmbH, Germany).

Proteins uniformly labelled by $^{15}N$ or $^{15}N{}^{13}C$ for NMR studies were overexpressed in *E. coli* BL21(DE3) (YqgP NTD) or *E. coli* Lemo21(DE3) (YqgP CTD), grown in M9 minimal medium containing partial M9 salt solution (6.8 g/l $Na_2HPO_4.2H_2O$, 3 g/l $KH_2PO_4$, pH 7.4, 0.5 g/l NaCl), 1 mM $MgSO_4$, 0.1 mM $CaCl_2$, 0.5% (w/v) D-glucose (U-$^{13}C_6$, 99%, Cambridge Isotope Laboratories) and 1 g/l $NH_4Cl$ ($^{15}N$, 99%, Cambridge Isotope Laboratories).

## Protein expression and purification

Proteins for structural and calorimetric studies were prepared by overexpression in *E. coli* BL21Star™(DE3) (Invitrogen) and Lemo21(DE3) (Wagner *et al*, 2008) (New England Biolabs) strains grown in LB medium. This included the C-terminally 6xHistidine-tagged TEV site-containing N-terminal extramembrane domain of YqgP [6xHis-TEVsite-YqgP(1–177); NTD], and the N-terminally 6xHistidine-tagged TEV site-containing C-terminal extramembrane domain of YqgP [YqgP(384–507)-TEVsite-6xHis; CTD] expressed from the pET25b or pHIS-2 vector, respectively. The N-terminal cytosolic extramembrane domain of MgtE [MgtE(2-275)] for antibody production was overexpressed in *E. coli* Lemo21(DE3) as a GST-6xHistidine-TEVsite-MgtE(2–275) construct from the pGEX6P1 vector.

His_6-YqgP(1–170) and His_6-YqgP(385–507) for antibody production were expressed in *E. coli* BL21(DE3)pLysS from pRSET-A vector and purified using metal-chelate affinity chromatography.

Typically, protein production was induced with 0.5 mM IPTG at 25°C with aeration and gentle shaking overnight. Bacterial cultures were harvested by centrifugation at 5,000 × *g* for 10 min at 4°C, and cells were resuspended in isolation buffer composed of 20 mM HEPES, pH 7.4, 300 mM NaCl, 10% (v/v) glycerol, 10 mM imidazole, 1 mM PMSF and 1 × cOmplete™ EDTA-free Protease Inhibitor Cocktail (Roche). Cells were lysed by three passages through the high-pressure homogeniser Emulsiflex®-C3 (Avestin, Inc.), and lysates were cleared from the cell debris by centrifugation (15,000 × *g*, 30 min, 4°C). Supernatants were loaded on Ni-NTA Agarose (Qiagen) equilibrated in isolation buffer, and His-tagged protein was eluted specifically with increasing concentration of imidazole in elution buffer (20 mM HEPES, pH 7.4, 300 mM NaCl, 10% (v/v) glycerol). The GST-tagged MgtE(2–275) domain was additionally purified using Glutathione Sepharose® 4 Fast Flow (Merck) and specifically eluted by 10 mM reduced glutathione into 20 mM HEPES, pH 7.4, 300 mM NaCl, 10% (v/v) glycerol. The 6xHistidine tags from YqgP domains and GST-6xHistidine tags from MgtE domain were removed by incubation with 6xHistidine-tagged TEV protease at protease-to-substrate ratio of ~ 1:200 (w/w), in 5 mM Tris–HCl, pH 8, 1 mM EDTA, 1 mM β-mercaptoethanol for 16 h at 25°C. The cleaved protein was then buffer-exchanged into isolation buffer (this paragraph) using PD-10 desalting column (GE Healthcare Life Sciences) and loaded onto Ni-NTA agarose. The flow-through fraction containing tag-free YqgP or MgtE domains was collected, concentrated by ultrafiltration (Vivaspin, Sartorius) and used for further applications.

## *In vitro* translation

Reference fragments of *B. subtilis* MgtE encompassing the first 300, 315, 330, 340, 355 and 370 residues of MgtE were generated by PCR amplification of the respective coding regions from *B. subtilis* 168 genomic DNA, and *in vitro*-transcribed and *in vitro*-translated as described previously (Lemberg & Martoglio, 2003; Strisovsky *et al*, 2009). Briefly, PCR was performed using set of *mgtE*-specific primers containing SP6 RNA polymerase promoter and ribosome-binding site in the forward primer, and stop codon in the reverse primer. Messenger RNAs were purified by LiCl precipitation and translated using Wheat Germ Extract (Promega). The resulting crude translation mixtures containing the MgtE reference fragments were separated using SDS–PAGE and visualised by polyclonal anti-MgtE(2–275) antibody and immunoblotting.

## Production of primary antibodies

Purified His_6-YqgP(1–170), His_6-YqgP(385–507) and MgtE(2–275) were used to raise rabbit polyclonal antibodies (Agro-Bio, France). Forty-two days after immunisation, the sera were collected and purified by affinity chromatography on an Affi-Gel 10 (Bio-Rad) column containing covalently immobilised antigen as described before (Campo & Rudner, 2006). Briefly, approximately 2 mg of purified antigen was dialysed into coupling buffer (20 mM HEPES pH 8, 200 mM NaCl, 10% (v/v) glycerol) and coupled to 1 ml Affi-Gel 10 resin (Bio-Rad) as described by the manufacturer. After coupling,

the antigen resin was washed with 100 mM glycine pH 2.5 and neutralised with 1× PBS to remove all uncoupled protein. Twenty ml of antisera were batch-adsorbed to the antigen resin overnight at 4°C. The resin was loaded into a 5-ml column and washed with 20 ml 1× PBS, 50 ml 1× PBS 500 mM NaCl and 10 ml 0.2× PBS. The affinity-purified antibodies were then eluted with 100 mM glycine pH 2.5. The peak fractions were pooled and dialysed with three changes into 1× PBS with 50% (v/v) glycerol and stored at −20°C.

## Immunoblotting

*Bacillus subtilis* cultures in exponential phase ($OD_{600}$ of ~ 1) were harvested by centrifugation (5,000 × *g*, 10 min, 25°C). Cell pellet was resuspended in lysis buffer containing 20 mM Tris–HCl, pH 7.5, 10 mM EDTA, 1 mM PMSF and 1× cOmplete™ EDTA-free Protease Inhibitor Cocktail (Roche), 1 mg/ml lysozyme and 2500 U Pierce Universal Nuclease (cat no. 88700, Thermo Fisher) and incubated at 37°C, for 10 min. Reducing SDS–PAGE loading buffer was added, and the mixture was heated to 60°C for 10 min with vigorous shaking. Samples were separated on Tris–glycine SDS–PAGE and electroblotted onto a nitrocellulose membrane (Serva). Equivalent total protein amounts were loaded as estimated by the $OD_{600}$ values, and after electrotransfer, total protein in each lane was quantified using the revert total protein stain (LI-COR®). The following antibodies were used for immunodetection: anti-thioredoxin antibody produced in rabbit (Merck), polyclonal anti-MgtE (this study), polyclonal anti-YqgP$_{NTD}$ (this study) and polyclonal anti-YqgP$_{CTD}$ (this study) antibodies all raised in rabbit (Agro-Bio, France), and rabbit polyclonal anti-FtsH (gift from Prof. Thomas Wiegert, Zittau, Germany). Donkey anti-rabbit IgG (H + L) Cross-Adsorbed Secondary Antibody DyLight 800 conjugate (Invitrogen) was used for visualisation by near-infrared laser scanning on the Odyssey® CLx Imaging System (LI-COR®) according to manufacturer's instructions. The resulting fluorescence intensity data for individual proteins were optionally used for quantification, as specified in each case. For occasional chemiluminescent detection (as noted in figure legends), goat anti-rabbit IgG conjugated to horseradish peroxidase (HRP) (Dako) was used as a secondary antibody, and HRP activity was detected by the Western Lightning Pro Kit (PerkinElmer).

## Rhomboid activity assay *in vivo*

*Bacillus subtilis* cells overexpressing particular YqgP variant under the control of $P_{xyl}$ promoter were cultivated as described above. The amounts of the uncleaved and cleaved forms of MgtE or model substrate TatA were measured by quantitative fluorescence immunoblotting as described in the previous section. Since the integrated intensity of fluorescence signal for a given band is proportional to the molarity of the antigen, substrate conversion at steady state was quantified as the ratio of fluorescence intensities of bands corresponding to the cleavage product (P) divided by the sum of signals for the full-length substrate (S) and the cleavage product (P), i.e. signal(P)/[signal(S) + signal(P)]. Relative "specific activities" at steady state *in vivo* were then calculated by dividing substrate conversion by the signal from the expressed rhomboid enzyme (YqgP variant) corrected for the total protein signal in that gel lane.

## SILAC-based quantitative proteomics and data evaluation

For SILAC-based quantitative proteomics, the BS50 and BS51 (Table EV2) strains of *B. subtilis* auxotrophic for lysine were cultivated in the medium described in section *Bacterial strains, media and growth conditions*. Equal amounts of heavy and light cell cultures (based on optical density at 600 nm, $OD_{600}$) were mixed at harvest and centrifuged. Cell pellets were then resuspended in lysis buffer containing 20 mM HEPES, pH 7.4, 100 mM NaCl, 10% (v/v) glycerol, 1 mM EDTA, 50 μg/ml lysozyme and 0.4× MS-Safe Protease and Phosphatase Inhibitor Cocktail (Merck) and incubated at 37°C, for 15 min and on ice for 15 min, afterwards. Cells were lysed by sonication using SONOPLUS HD2200 device (BANDELIN) set to 4 × 30 s pulse cycle and 30% power. Cell debris was removed by centrifugation (15,000 × *g*, 30 min, 4°C). Crude membranes were isolated by ultracentrifugation (100,000 × *g*, 1.5 h, 4°C) and washed sequentially with 0.1 M $Na_2CO_3$ and 1 M NaCl.

*Bacillus subtilis* transmembrane protein-enriched fractions were separated on 4–20% gradient Tris–glycine SDS–PAGE system (Bio-Rad). Two SILAC experiments with swapped labelling were conducted: Experiment 1 [marked "heavy-to-light" or [H/L]]: heavy BS50 (YqgP), light BS51 (YqgP.S288A); and Experiment 2 [marked "light-to-heavy" or [L/H]]: light BS50, heavy BS51. Both experiments were resolved in separate gel lanes, which were subsequently sliced into five fractions (A–E) each (Fig 1), and individually digested using a standard in-gel digestion protocol. Briefly, every gel slice was first destained with 25 mM ammonium bicarbonate in 50% (v/v) acetonitrile (ACN, Merck), dehydrated and shrunk using acetonitrile (ACN) in a reducing environment of 5 mM 1,4-dithiothreitol (DTT, Merck) and incubated for 30 min at 65°C. Fractions were then alkylated using 12.5 mM iodoacetamide (Merck), shrunk and digested using 0.1 μg trypsin at 37°C overnight. Tryptic peptides were extracted with 60% (v/v) ACN in 0.5% (v/v) trifluoroacetic acid (TFA, Merck), dried and reconstituted in 20 μl of 2% (v/v) ACN with 0.1% (v/v) formic acid (FA, Merck) and analysed via LC-MS/MS.

The LC-MS/MS analyses were performed using UltiMate 3000 RSLCnano System (Dionex) coupled to a TripleTOF 5600 mass spectrometer with a NanoSpray III source (AB Sciex). After injection, the peptides were trapped and desalted in 5% (v/v) ACN/0.1% (v/v) FA at a flow rate of 5 μl/min on an Acclaim® PepMap 100 column (5 μm, 2 cm × 100 μm ID, Thermo Scientific) for 5 min. The separation of peptides was performed on an Acclaim® PepMap 100 analytical column (3 μm, 25 cm × 75 μm ID, Thermo Scientific) using a gradient from 5% (v/v) to 18% (v/v) ACN, over 12 min, with a subsequent rise to 95% (v/v) ACN/0.1% (v/v) FA, over 30 min. TOF MS scans were recorded from 350 to 1,250 m/z, and up to 18 candidate ions per cycle were subjected to fragmentation. Dynamic exclusion was set to 10 s after one occurrence. In MS/MS mode, the fragmentation spectra were acquired within the mass range of 100–1,600 m/z.

The quantitative mass spectrometric data files were processed and analysed using MaxQuant (v1.5.2.8) (Cox & Mann, 2008). The search was performed using a UniProt/Swiss-Prot *B. subtilis* database (downloaded 17/05/15) with common contaminants included. Enzyme specificity was set to trypsin, with methionine oxidation as a variable modification. Cysteine carbamidomethylation was considered as a fixed modification. The heavy SILAC label was set to K8, the minimal peptide length to 6, and two missed cleavages were

allowed. Proteins were considered as identified if they had at least one unique peptide, and quantified if they had at least one quantifiable SILAC pair. Transmembrane topology predictions were obtained using Phobius (Käll *et al*, 2004), and additional sequence visualisations were obtained using QARIP software (Ivankov *et al*, 2013; Fig 1).

## Quantitative proteomics for interactor identification

All procedures were performed essentially as described (Doan *et al*, 2009). For crude membrane preparation, cells were grown in LB medium, at 37°C. At $OD_{600}$ of 0.6, 50 ml of culture was harvested for membrane preparation. Membrane proteins were solubilised by the addition of the detergent NP-40 to a final concentration of 1% (v/v). The soluble fraction was mixed with 25 μl anti-GFP antibody resin (Chromotek, Germany) and rotated for 3 h at 4°C. The resin was pelleted at $3,000 \times g$, and the supernatant was removed. After washes, immunoprecipitated proteins were eluted by the addition of 85.5 μl of SDS–PAGE sample buffer and heated for 15 min at 50°C. The eluted material (the IP) was transferred to a fresh tube, and 2-mercaptoethanol was added to a final concentration of 10% (v/v).

Proteins extracted from the co-immunoprecipitation eluates were resolved using NuPAGE 4–12% system (Thermo Fisher Scientific), stained with Coomassie blue (R250, Bio-Rad) and in-gel digested using modified trypsin (Promega, sequencing grade) as described, previously (Casabona *et al*, 2013). Of note, samples from replicates 1 and 2 were prepared using oxidation and reduction–alkylation procedures as described (Jaquinod *et al*, 2012). The peptides were analysed by online NanoLC-MS/MS (UltiMate 3000 and LTQ Orbitrap Velos, Thermo Scientific). In particular, peptides were loaded on a 300 μm × 5 mm PepMap C18 precolumn (Thermo Scientific) and separated on a homemade 75 μm × 150 mm C18 column (Gemini C18, Phenomenex). MS and MS/MS data were acquired using Xcalibur (Thermo Scientific). Mascot Distiller (Matrix Science) was used to produce mgf files before identification of peptides and proteins using Mascot (version 2.6) through concomitant searches against UniProt (*B. subtilis* strain 168 taxonomy, September 2018 version), GFP-tagged YqgP sequences, classical contaminants database (homemade) and the corresponding reversed databases. The Proline software (http://proline.profiproteomics.fr) was used to filter the results (conservation of rank 1 peptides, peptide identification FDR < 1% as calculated on peptide scores by employing the reverse database strategy, minimum peptide score of 25 and minimum of 1 specific peptide per identified protein group) before performing a compilation, grouping and comparison of the protein groups from the different samples. Proteins were considered as potential partners of the bait if they were identified only in the positive co-IPs with a minimum of three weighted spectral counts or enriched at least five times in positive co-IPs compared with control ones based on weighted spectral counts.

## Isothermal titration calorimetry

The binding of divalent cations to the extramembrane domains of YqgP (NTD, CTD) was investigated using titration microcalorimetry. All titration experiments were performed in 50 mM Tris–HCl, pH 7.3, 150 mM NaCl at 25°C using Auto-iTC$_{200}$ instrumentation (MicroCal, Malvern Panalytical Ltd). Prior to the ITC measurements, protein samples were dialysed into the same buffer in which

divalent cation salt solutions were prepared. Typically, 200 μl of the protein sample was titrated by stepwise 2 μl injections of divalent cation solution. Protein and ligand concentrations were adjusted as follows: 1.5 mM NTD was titrated by 50 mM $MnCl_2$, 50 mM $MgSO_4$, 50 mM $ZnCl_2$ or 50 mM $CaCl_2$, 0.7 mM NTD by 15 mM $CoCl_2$ and 0.25 mM NTD by 7 mM $NiCl_2$-buffered solutions. Control experiments included ligand titration into the buffer only. Concentrations of stock solutions of the analysed proteins were determined by quantitative amino acid analysis, and concentrations of the stock solutions of divalent metal cation salts were determined by elemental analysis. Titration data were processed and analysed using MicroCal Origin 7.0 (MicroCal, Malvern Panalytical Ltd).

## Nuclear magnetic resonance spectroscopy

For structure determination of YqgP N-terminal domain (YqgP$_{NTD}$), 350 μl samples of 1 mM $^{15}N/^{13}C$ uniformly labelled protein in 25 mM phosphate buffer pH 6.5 containing 150 mM NaCl and 5% $D_2O/95\%$ $H_2O$ were used. All NMR spectra for backbone assignments and three-dimensional structure determination were collected on 850 MHz Bruker Avance spectrometer equipped with triple resonance ($^{15}N/^{13}C/^1H$) cryoprobe at 37°C. In a series of double and triple resonance, spectra (Renshaw *et al*, 2004; Veverka *et al*, 2006) were recorded to determine essentially complete sequence-specific resonance backbone and side-chain assignments. Constraints for $^1H$ $^1H$ distance required to calculate the structure of YqgP$_{NTD}$ were derived from 3D $^{15}N/^1H$ NOESY HSQC and $^{13}C/^1H$ NOESY HMQC, which were acquired using a NOE mixing time of 100 ms. The family of converged structures for YqgP$_{NTD}$ was initially calculated using Dyana 2.1 (Herrmann *et al*, 2002). The combined automated NOE assignment and structure determination protocol was used to automatically assign the NOE cross-peaks identified in the NOESY spectra and to produce preliminary structures. In addition, backbone torsion angle constraints, generated from assigned chemical shifts using the TALOS+ program (Shen *et al*, 2009), were included in the calculations. Subsequently, five cycles of simulated annealing combined with redundant dihedral angle constraints were used to produce a set of converged structures with no significant restraint violations (distance and van der Waals violations < 0.2 Å and dihedral angle constraint violation < 5°), which was further refined in explicit solvent using the YASARA software with the YASARA force field (Harjes *et al*, 2006). The structures with the lowest total energy were selected and analysed using the Protein Structure Validation Software suite (www.nesg.org). Cation-binding experiments were performed in 50 mM Tris–HCl, pH 7.3, 150 mM NaCl, 5% $D_2O$, at 37°C, when the same aliquots of the ligand were added to the protein solution stepwise, after every titration measurement.

# Data availability

The mass spectrometry (MS) proteomic data have been deposited to the ProteomeXchange Consortium (http://proteomecentral.proteomexchange.org) (Deutsch *et al*, 2017) via the PRIDE (Perez-Riverol *et al*, 2019) partner repository with the dataset identifiers PXD014578 (http://www.ebi.ac.uk/pride/archive/projects/PXD014578) and PXD014566 (http://www.ebi.ac.uk/pride/archive/projects/

PXD014566). The NMR structural data have been deposited in the Protein Data Bank (PDB) under accession code 6R0J (http://www.rcsb.org/pdb/explore/explore.do?structureId=6R0J) and in the Biological Magnetic Resonance Data Bank (BMRB) under the accession code 34376 (http://www.bmrb.wisc.edu/data_library/summary/index.php?bmrbId=34376).

**Expanded View** for this article is available online.

## Acknowledgements

We thank Libor Krásný for critical reading of the manuscript. TD thanks David Rudner and Eric Cascales for their support and guidance. We thank Radko Souček for amino acid analysis and Stanislava Matějková for elemental analysis. KS acknowledges support from Czech Science Foundation (project no. 18-09556S), and KS and MK acknowledge financial support from the Ministry of Education, Youth and Sports of the Czech Republic (project no. LO1302), European Regional Development Fund (ERDF/ESF) (project No. CZ.02.1.01/0.0/0.0/16_019/0000729), Gilead Sciences & IOCB Research Centre, and the National Subvention for Development of Research Organisations (RVO: 61388963) to the Institute of Organic Chemistry and Biochemistry. This work was also supported by the CNRS, Aix-Marseille University (AMU) and a Marie-Curie International Reintegration Grant (PIRG08-GA-2010-276750) to TD. BC and JD were supported by fellowships from AMU, and JD also by the Fondation pour la Recherche Médicale (FRM, FDT20160435133). Proteomic experiments were partly supported by the Agence Nationale de la Recherche (ProFI grant ANR-10-INBS-08-01). JB and Jana Březinová acknowledge support from the Grant Agency of Charles University (GA UK) in Prague (PhD grant project no. 170214).

## Author contributions

KS, TD, BC and JBe designed research; JBe, BC, TD, VV, RH, PS, MK, JD, MB, YC and PR performed research and analysed data; and KS and JBe wrote the manuscript with contributions from TD, JBř, BC, VV, AG, RH, PS, YC and MK.

## Conflict of interest

The authors declare that they have no conflict of interest.

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
