## [Review Process File · The EMBO Journal]

Rhomboid intramembrane protease YqgP licenses bacterial membrane protein quality control as adaptor of FtsH AAA protease

Jakub Began, Baptiste Cordier, Jana Březinová, Jordan Delisle, Rozálie Hexnerová, Pavel Srb, Petra Rampírová, Milan Kožíšek, Mathieu Baudet, Yohann Couté, Anne Galinier, Václav Veverka, Thierry Doan and Kvido Strisovsky.

Review timeline:

Submission date:	12 th July 2019
Editorial Decision:	29 th August 2019
Revision received:	18 th November 2019
Accepted:	12 th December 2019

Editor: Hartmut Vodermaier

Transaction Report:

1st Editorial Decision

29th August 2019

Thank you again for submitting your manuscript on bacterial rhomboid links to AAA proteases and membrane protein quality control for our editorial consideration. We have now finally received a complete set of comments from three expert referees, who had kindly taken the time to also review the co-submitted manuscript. I am happy to inform you that all three referees are generally supportive and acknowledge both the interest and the technical quality of the work. I would therefore like to invite you to revise the manuscript in response to the comments enclosed below.

REFERE REPORTS

Referee #1:

This study has investigated the substrates and function of the rhomboid family protease YqgP in *B. subtilis*. Two important discoveries are reported. First, the multipass membrane protein MgtE is a bona fide endogenous substrate for YqgP. Second, YqgP also engages MgtE and its cleavage products in a non-proteolytic role for delivery to FtsH, a protease that mediates MgtE degradation. The importance of these events for the physiology of *B. subtilis* is illustrated via growth assays under heavy metal toxicity conditions. These experiments suggest that YqgP is acting to limit MgtE levels particularly under conditions of non-Mg²⁺ heavy metal excess to limit their toxicity. Mechanistic studies identify the N-terminal domain as participating in metal sensing (particularly Mn²⁺), including structural analysis that identifies a putative Mn²⁺ binding site. The study is important, interesting, and exciting. I am strongly supportive of publication in EMBO J. pending a few clarifications and improvement of the quality of a few pieces of key data.

Major points:

1) The phenotype seen in Fig. 3B with $\Delta YqgP$ is an important result, but the effect is relatively modest. For this reason, it is worth stating in the text that this effect is highly reproducible (assuming that it is, given the similar result in Fig. 3C) or better yet, putting error bars on the graph from the average of multiple independent experiments.

2) The effect in Fig. 4B appears to be very similar for all three YqgP constructs (wild type, ΔNTD , ΔCTD) when one inspects the cleavage blots. It seems that the difference only emerges in the quantification when one 'normalizes' for the YqgP levels (bottom blots), which seem to be preferentially elevated only at the highest Mn^{2+} level and only for the ΔNTD construct. As this is the ONLY point where a difference is seen (and only with normalization), I am not entirely convinced of the validity of the result. Replicates of this important experiment with error bars would make it more convincing. Also, can the authors please explain why only the ΔNTD preferentially increases with 100 $\mu M Mn^{2+}$.

3) The difference between lanes 13-16 versus 37-40 intended to show that FtsH contributes to degradation are not so convincing (at least not to me) by inspecting the gel. Perhaps quantification of results from multiple experiments would better illustrate this relatively subtle effect.

Minor points:

1) The authors should clarify in the text that the proteomic analysis for substrates relied on detecting peptides for a protein at a lower-than-expected region of the gel selectively in the YqgP strain relative to the S288A mutant. This is currently only evident in the figure legend.

2) In Fig. 3C, is the small growth difference seen in the bottom panel reproducible, and if so, what is the basis of this effect?

3) The authors should comment on the observation that re-expression of YqgP does not merely restore the wild type phenotype but is substantially protective. I presume it is due to over-expression? Regardless, the observation merits explanation in the text.

4) Fig. 4A should be interpreted with caution. "Normalizing" the cleavage by the amount of YqgP based on western blots is highly unreliable because detection efficiencies can vary widely when comparing variants of proteins that migrate at different positions on the gel. I feel the only really solid conclusion that can be drawn is that the ΔNTD construct has a preferential effect on MgtE, and the ΔCTD might lead to increased cleavage (but might equally have little or no effect). Since the second conclusion regarding ΔCTD is not important to the study, the authors gain little by over-interpreting this result. I recommend softening the conclusions regarding the CTD.

5) Fig. 7E is not really a "pulse-chase" experiment as described. It is better described as a translation shut-off chase. Pulse-chase would involve radiolabelling for a 'pulse' to monitor only the proteins made during that time. Please re-phrase.

6) The Lilley and Ploegh (2004) citation for SPP acting in dislocation in the absence of activity seems to be incorrect. Perhaps the authors meant to cite Loureiro et al., 2006, Nature.

7) The authors should discuss the possible relationship between YqgP and ATP synthase in the Discussion section. This was a major interactor, and seems to be specific, so a reader may appreciate some thoughts about what it might mean.

Referee #2:

The paper by Jakub Began and co-workers describes a new regulatory role for the intramembrane protease GlpG homologue from *Bacillus subtilis* YqgP.

The authors initially used quantitative proteomics and co-IP MS/MS to show that the GlpG homologue YqgP could be interacting with the magnesium transporter MgtE. Using an antibody generated against the N-terminal domain of MgtE they could show that both endogenous and ectopically expressed YqgP cleaved MgtE and this cleavage could be reduced by a rhomboid-

specific inhibitor. They then screened for conditions that would increase catalytic activity of MgtE by YqgP and, indeed, found increases proteolysis under low Mg²⁺ and high Mn²⁺ conditions, which has physiological relevance.

Having shown a correlation between MgtE and YqgP, then could show its physiological relevance by *B. subtilis* growth assays in which Mn²⁺ was toxic to growth in an YqgP deletion strain, but could be rescued with complementation of YqgP (and a catalytically inactive mutant). They were next able to show that the MgtE mediated degradation by YqgP was facilitated by an N-terminal domain in the protein that bound Mn²⁺ (but not Mg²⁺) and further determined the structure of the N-terminal domain by NMR. They identified residues coordinating Mn²⁺ and alanine mutations of these residues were just as effective as an N-terminal function mutant of YqgP at reducing MgtE degradation by YqgP. Intriguingly, they further concluded that the direct interaction with the N-terminal domain of YqgP and MgtE was sufficient to promote degradation in the absence of active protease as it recruited FtsH (the zinc metalloprotease). In either the absence of YqgP, or in its presence but with the rhomboid inhibitor added, FtsH was not able to degrade MgtE at all. The authors concluded a model where YqgP cleaves MgtE under low Mg and high Mn²⁺ concentrations that further facilitates the protein's degradation by the general protease FtsH.

This tour-de-force study was a pleasure to read and unveils a new line of enquiry into the enigmatic physiological function of GlpG and their homologues. Whilst it may seem "odd" at first glance that a protease would evolve for such a specific set of conditions, in actual fact, transporters are known to have finely-tuned regulation. For example, some amino acid transporters belonging to the MFS require an specialised ER chaperone (Shr3) to aid in their folding (J Cell Biol. 2007 Feb 26; 176(5): 617-628.); this activity is also associated with ERAD. The same transporter family is subject to down regulation under high nutrient conditions, by an increase in ubiquitin ligase activity (Rsp5), that is a signal for their trafficking from the plasmembrane to multi-vesicular bodies (Trends Cell Biol. 2010 Apr;20(4):196-204). Thus, there is a precedence of specialised machinery to cope with transporters which have "house-keeping" roles in the cell.

Major concerns:

Technically I find this paper excellent. I have 1 major comment.

1. It seems that the rhomboid proteases have developed a function for a unique set of conditions. Part of this conclusion hangs on the physiological relevance of MgtE mediated-degradation under low Mg²⁺ and high Mn²⁺ conditions, which the authors describe in the discussion as having relevance in the habitat of the soil in which the gram-positive bacterium grows. The only "fly-in-the-ointment" is that after the initial toxicity to growth upon the addition of MnSO₄ at 100 mins (Fig. 3B), the cell growth seems to be recovering again at 300 mins. What is the OD₆₀₀ after 10 hours? Do the cells recover from this phenotype and, if so, why? Is it possible that the bacterium switches on other compensation pathways to avoid Mn²⁺ toxicity by MgtE? Using your MgtE specific antibody do you see a reduction in MgtE expression at longer incubation times?

2. It is mentioned in the introduction that *B. subtilis* has another GlpG homologue (YdcA), but this homologue is not catalytically active. But given that the catalytically inactive mutant of YqgP can interact with FtsH, I was curious if the YdcA protein had a similar NTD domain? If so, might it be possible that any recovery in growth with MnSO₄ might be related to YdcA, which is unregulated in the absence of YqgP and also have pseudoprotease activity?

Minor comments:

1. I am curious about the recognition of YqgP for MgtE. Using the delta G predictor by von Heine's lab (<http://dgpred.cbr.su.se/index.php?p=home>) it seems the second TM is not so favourable for insertion and I presume that it is driven to insert by TM1, which makes sense given the short loop between TM1 and TM2 segments. Since Mg²⁺ is coordinated directly by TM2 (Nature Communications volume 5, Article number: 5374 (2014)), I guess its possible that MgtE may not form a stable fold with low Mg²⁺ and therefore becomes more susceptible to degradation? I was curious if you had analysed MgtE mutants that abolish Mg²⁺ coordination by TM2?

Referee #3:

Despite detailed structural insights, very little is known about the cognate substrates and biological functions of rhomboid proteases in bacteria. Two co-submitted manuscripts now fill this gap, one identifying substrates of two rhomboids in the Gram-negative pathogen *Shigella sonnei* and one revealing a rhomboid function in the Gram-positive model bacterium *Bacillus subtilis*. Both manuscripts present exciting results and are valuable contributions to the field. Since they report strikingly different substrates (orphan single-pass transmembrane proteins versus a multi-pass membrane transporter) and biological functions, the discussions would benefit from cross-referencing and comparing the results in case both papers appear back-to-back.

Began et al. used quantitative proteomics approaches to identify substrates and interactors of the *Bacillus* rhomboid protease YqgP. They found that YqgP cleaves the magnesium transporter MgtE under conditions when it imports toxic ions (low magnesium, high manganese or zinc) and that it interacts with the FtsH protease. Interestingly, YqgP is a metal sensor and binds manganese and zinc ions via its N-terminal domain. Furthermore, YqgP delivers MgtE to FtsH for full degradation. The authors use a wide range of biochemical and biophysical techniques and conclude with interesting evolutionary considerations. I only have a few minor questions and comments.

1. Fig. 1E: Might the strikingly high number of ATPase subunits interacting with YqgP be of physiological significance?
2. Are there sequence similarities between the cleavage sites in MgtE and TatA?
3. In the paragraph describing Fig. 7E (page 15 of the PDF), the authors state that FtsH was not able to act on MgtE at all and refer to lines 25-32. Shouldn't this be lines 1-8? In lines 25-32 both proteases are absent.
4. Page and/or line numbers would facilitate reviewing.

1st Revision - authors' response

18th November 2019

Responses to referees' comments are shown in blue below.

Referee #1:

This study has investigated the substrates and function of the rhomboid family protease YqgP in *B. subtilis*. Two important discoveries are reported. First, the multipass membrane protein MgtE is a bona fide endogenous substrate for YqgP. Second, YqgP also engages MgtE and its cleavage products in a non-proteolytic role for delivery to FtsH, a protease that mediates MgtE degradation. The importance of these events for the physiology of *B. subtilis* is illustrated via growth assays under heavy metal toxicity conditions. These experiments suggest that YqgP is acting to limit MgtE levels particularly under conditions of non-Mg²⁺ heavy metal excess to limit their toxicity. Mechanistic studies identify the N-terminal domain as participating in metal sensing (particularly Mn²⁺), including structural analysis that identifies a putative Mn²⁺ binding site. The study is important, interesting, and exciting. I am strongly supportive of publication in EMBO J. pending a few clarifications and improvement of the quality of a few pieces of key

data.

Major points:

1) The phenotype seen in Fig. 3B with $\Delta YqgP$ is an important result, but the effect is relatively modest. For this reason, it is worth stating in the text that this effect is highly reproducible (assuming that it is, given the similar result in Fig. 3C) or better yet, putting error bars on the graph from the average of multiple independent experiments.

Indeed, the effect is highly reproducible, as indicated by similar independent results shown in Fig. 3B, 3C and 5C. Nevertheless, we have repeated the experiment from Fig. 3B in biological triplicates, and now show growth all datapoints overlaid by lines connected mean values. We hope that the tightness of the data in the new Fig. 3B allows us to show representative measurements in other figures reporting similar experiments (Fig. 3CDE, 5C).

2) The effect in Fig. 4B appears to be very similar for all three YqgP constructs (wild type, ΔNTD , ΔCTD) when one inspects the cleavage blots. It seems that the difference only emerges in the quantification when one 'normalizes' for the YqgP levels (bottom blots), which seem to be preferentially elevated only at the highest Mn²⁺ level and only for the ΔNTD construct. As this is the ONLY point where a difference is seen (and only with normalization), I am not entirely convinced of the validity of the result. Replicates of this important experiment with error bars would make it more convincing. Also, can the authors please explain why only the ΔNTD preferentially increases with 100 μM Mn²⁺.

We acknowledge that it is important that the differences be evident also from the blot to the human eye and not rely solely on densitometric quantification. We have re-analyzed the samples shown in western blots of Fig. 4B and show a technically better version of this blot where differences are more evident above background. The apparent increase in the intensity of the ΔNTD band at 100 μM Mn²⁺ is caused by a difference in loading (which we had corrected for during quantification), as is documented below by raw data for this blot (total protein stain and anti-CTD blot signals).

Densitometry by near-infrared fluorescence scanning:

Condition	YggP Signal	total protein	YggP signal (normalized)	Fold signal (Mn1=1)
ΔNTD + 1 μM MnCl ₂	36726,00	1391703,84	0,0264	1,00
ΔNTD + 20 μM MnCl ₂	39338,21	1349046,65	0,0292	1,10
ΔNTD + 100 μM MnCl ₂	61953,78	2074249,71	0,0299	1,13

Importantly, quantification of the fluorescence of the new and original blot gives virtually identical results. To back up this important result statistically, we have now replicated the experiment four times and plot quantification results into a bar graph as fold MgtE cleavage activation by 100 μM MnCl₂ compared to 1 μM MnCl₂ as average value and four data points per construct (Fig. 4D), hopefully making the data more convincing.

3) The difference between lanes 13-16 versus 37-40 intended to show that FtsH contributes to degradation are not so convincing (at least not to me) by inspecting the gel. Perhaps quantification of results from multiple experiments would better illustrate this relatively subtle effect.

We have now done the experiment covering lanes 13-16 versus 37-40 in triplicate and show now a new Fig. EV2 displaying that in the presence of FtsH, the depletion of the cleavage products of MgtE is faster than in the absence of FtsH, consistently illustrating the effects that we describe in text and show on the western blot in Fig. 7E.

Minor points:

1) The authors should clarify in the text that the proteomic analysis for substrates relied on detecting peptides for a protein at a lower-than-expected region of the gel selectively in the YggP strain relative to the S288A mutant. This is currently only evident in the figure legend.

We have now made this clearer by stating “We then conducted quantitative proteomics analysis of membrane fractions of both of these strains employing SILAC labelling and SDS PAGE pre-fractionation (GeLC experiment), where we were seeking peptides belonging to proteins migrating at a lower-than-expected apparent molecular weight selectively in the YqgP-expressing strain relative to the S288A mutant expressing one.”, on page 11.

2) In Fig. 3C, is the small growth difference seen in the bottom panel reproducible, and if so, what is the basis of this effect?

Although intriguing at first sight, after careful analysis we state that this effect is within the experimental error of the measurement and thus not significant, and update Fig. 3C accordingly.

3) The authors should comment on the observation that re-expression of YqgP does not merely restore the wild type phenotype but is substantially protective. I presume it is due to over-expression? Regardless, the observation merits explanation in the text.

Indeed, this is due to overexpression of YqgP. To illustrate this point we now add a new panel D into Fig. 3, which shows that the increase in fitness due to YqgP can be decreased by a YqgP-specific inhibitor in a dose-dependent manner, both in an overexpression situation and with endogenous YqgP.

4) Fig. 4A should be interpreted with caution. "Normalizing" the cleavage by the amount of YqgP based on western blots is highly unreliable because detection efficiencies can vary widely when comparing variants of proteins that migrate at different positions on the gel. I feel the only really solid conclusion that can be drawn is that the Δ NTD construct has a preferential effect on MgtE, and the Δ CTD might lead to increased cleavage (but might equally have little or no effect). Since the second conclusion regarding Δ CTD is not important to the study, the authors gain little by over-interpreting this result. I recommend softening the conclusions regarding the CTD.

We agree with this reviewer and tone down the statement on Δ CTD in the text. In addition, we now emphasise the effect of Δ NTD and de-emphasise that of Δ CTD also in the bar graphs of Fig. 4B by using a broken Y axis.

5) Fig. 7E is not really a "pulse-chase" experiment as described. It is better described as a translation shut-off chase. Pulse-chase would involve radiolabelling for a 'pulse' to monitor only the proteins made during that time. Please re-phrase.

We apologize for confusing nomenclature. We now strictly name this experiment a 'translation shut-off' (chase) experiment.

6) The Lilley and Ploegh (2004) citation for SPP acting in dislocation in the absence of activity seems to be incorrect. Perhaps the authors meant to cite Loureiro et al., 2006, Nature.

We thank for the notice and apologize for a mental frameshift on this reference. Now duly corrected as suggested.

7) The authors should discuss the possible relationship between YqgP and ATP synthase in the Discussion section. This was a major interactor, and seems to be specific, so a reader may appreciate some thoughts about what it might mean.

We thank this reviewer for the note. We omitted the discussion of ATP synthase connection due to perceived space limitation, but now are glad to be able to add it back to the Discussion section on pages 18 and 19:

" Intriguingly, in addition to the role of YqgP in regulating MgtE, we suspect it can be engaged in other functions. In co-immunoprecipitation experiments with YqgP, we identified a number of ATP synthase subunits with high confidence and abundance by mass spectrometry (Fig. 1E). Since we have not identified any of these subunits in our substrate screen (Fig. 1C), we did not follow them up. However, this finding raises the possibility that further to its role in MgtE proteostasis, YqgP may be involved in the relationship between magnesium homeostasis and ATP synthesis, which would be consistent with a growing body of evidence supporting a regulatory relationship between magnesium and ATP levels. First, it was reported that a magnesium transporter-defective strain of *B. subtilis* can be rescued by inactivation of ATP synthase encoding genes (Wakeman et al., 2014), suggesting that lowering ATP levels allows the cell to cope with magnesium deprivation. This is consistent with findings from Groisman and colleagues in *Salmonella enterica*, where low levels of Mg²⁺ in the cell lead to ATP synthase inhibition and reduced ribosome biogenesis. As a result, the pool of magnesium that is not ATP-bound anymore, can be employed to support low levels of ribosome assembly, a priority process that requires large amounts of magnesium (Pontes et al., 2016). Finally, in the same bacterium, ATP synthase activity can be directly inhibited by the MgtC protein, whose expression is upregulated during magnesium deprivation (Lee, Pontes et al., 2013). Strikingly, MgtC downregulation in the membrane of *S. enterica* is somewhat reminiscent of how MgtE is controlled by YqgP and FtsH in *B. subtilis*. MgtR, a small hydrophobic peptide, is thought to destabilize MgtC prior to degradation by FtsH (Alix & Blanc-Potard, 2008). Hence, during magnesium deprivation and cation toxicity in *B. subtilis*, it is tempting to speculate that YqgP may also inhibit the activity of ATP synthase to preserve the free Mg²⁺ pool for translation purposes while assisting FtsH for the degradation of MgtE."

Referee #2:

The paper by Jakub Began and co-workers describes a new regulatory role for the intramembrane protease GlpG homologue from *Bacillus subtilis* YqgP.

The authors initially used quantitative proteomics and co-IP MS/MS to show that the GlpG homologue YqgP could be interacting with the magnesium transporter MgtE. Using an antibody generated against the N-terminal domain of MgtE they could show that both endogenous and ectopically expressed YqgP cleaved MgtE and this

cleavage could be reduced by a rhomboid-specific inhibitor. They then screened for conditions that would increase catalytic activity of MgtE by YqgP and, indeed, found increases proteolysis under low Mg^{2+} and high Mn^{2+} conditions, which has physiological relevance.

Having shown a correlation between MgtE and YqgP, they then could show its physiological relevance by *B. subtilis* growth assays in which Mn^{2+} was toxic to growth in an YqgP deletion strain, but could be rescued with complementation of YqgP (and a catalytically inactive mutant). They were next able to show that the MgtE mediated degradation by YqgP was facilitated by an N-terminal domain in the protein that bound Mn^{2+} (but not Mg^{2+}) and further determined the structure of the N-terminal domain by NMR. They identified residues coordinating Mn^{2+} and alanine mutations of these residues were just as effective as an N-terminal function mutant of YqgP at reducing MgtE degradation by YqgP. Intriguingly, they further concluded that the direct interaction with the N-terminal domain of YqgP and MgtE was sufficient to promote degradation in the absence of active protease as it recruited FtsH (the zinc metalloprotease). In either the absence of YqgP, or in its presence but with the rhomboid inhibitor added, FtsH was not able to degrade MgtE at all. The authors concluded a model where YqgP cleaves MgtE under low Mg and high Mn^{2+} concentrations that further facilitates the protein's degradation by the general protease FtsH.

This tour-de-force study was a pleasure to read and unveils a new line of enquiry into the enigmatic physiological function of GlpG and their homologues. Whilst it may seem "odd" at first glance that a protease would evolve for such a specific set of conditions, in actual fact, transporters are known to have finely-tuned regulation. For example, some amino acid transporters belonging to the MFS require a specialised ER chaperone (Shr3) to assist in their folding (J Cell Biol. 2007 Feb 26; 176(5): 617-628.); this activity is also associated with ERAD. The same transporter family is subject to down regulation under high nutrient conditions, by an increase in ubiquitin ligase activity (Rsp5), that is a signal for their trafficking from the plasmembrane to multi-vesicular bodies (Trends Cell Biol. 2010 Apr;20(4):196-204). Thus, there is a precedence of specialised machinery to cope with transporters which have "house-keeping" roles in the cell.

These are valid points and we now mention these examples in the manuscript page 18: "The specialised role that YqgP plays in the quality control of an essential transporter protein may seem unexpected, but conceptually, transporters are indeed regulated at a number of levels and very finely using specialised mechanisms. For example, amino acid permeases of the major facilitator superfamily in yeast are down-regulated from the cell surface under high nutrient conditions by an increase in ubiquitin E3 ligase activity of Rsp5, which initiates their trafficking from the plasma membrane to multi-vesicular bodies (Lauwers, Erpapazoglou et al., 2010). Other member of the same transporter family in yeast requires a specialised membrane chaperone Shr3, which assists the folding of the transporters and prevents their efficient degradation by ERAD (Kota, Gilstring et al., 2007)."

Major concerns:

Technically I find this paper excellent. I have 1 major comment.

1. It seems that the rhomboid proteases have developed a function for a unique set of conditions. Part of this conclusion hangs on the physiological relevance of MgtE mediated-degradation under low Mg²⁺ and high Mn²⁺ conditions, which the authors describe in the discussion as having relevance in the habitat of the soil in which the gram-positive bacterium grows. The only "fly-in-the-ointment" is that after the initial toxicity to growth upon the addition of MnSO₄ at 100 mins (Fig. 3B), the cell growth seems to be recovering again at 300 mins. What is the OD₆₀₀ after 10 hours? Do the cells recover from this phenotype and, if so, why? Is it possible that the bacterium switches on other compensation pathways to avoid Mn²⁺ toxicity by MgtE? Using your MgtE specific antibody do you see a reduction in MgtE expression at longer incubation times?

We thank this referee for the relevant points raised. To address them, we repeated the experiment and measured OD over a long period (see Fig. 3B). All strains recovered from initial toxicity but were not able to grow to the same OD as untreated cells during the observed time-scale, indicating that absence of YqgP indeed impacts on the fitness of *B.subtilis* under Mn stress at low Mg. We have not analysed the causes of this 'recovery' in detail, but we have not observed a reduction in MgtE expression at longer incubation times (see the figure below and please note higher activity of YqgP at later time-points under Mn stress). We can however hypothesise about the causes of the recovery based on the described genetic mechanisms related to the regulation of Mn metabolism. It is known that *B. subtilis* responds to Mn²⁺ excess in two ways. First, the MntR regulator senses excess intracellular Mn²⁺. It then represses the transcription of two Mn²⁺-uptake systems (MntABCD and MntH) and activates the expression of efflux systems (MneP and MneS) (Que and Helmann 2000 Mol Microbiol; Huang et al., 2016 Mol Microbiol). In addition, the two manganese efflux pumps, MneP and MneS, are also activated via Mn²⁺-responsive riboswitches (Dambach et al., 2015 Mol Cell; Price et al., 2015 Mol Cell). These actions take place at the transcriptional level, hence take some time to efficiently allow the cell to recover from an initial Mn²⁺ shock. As intracellular Mn²⁺ returns to physiological levels, cells resume growth and the role of YqgP thus apparently loses its importance for fitness in the later stages of the experiment. We have now incorporated this argument into Discussion on p. 17.

2. It is mentioned in the introduction that *B. subtilis* has another GlpG homologue (YdcA), but this homologue is not catalytically active. But given that the catalytically inactive mutant of YggP can interact with FtsH, I was curious if the YdcA protein had a similar NTD domain? If so, might it be possible that any recovery in growth with MnSO₄ might be related to YdcA, which is unregulated in the absence of YggP and also have pseudoprotease activity?

Indeed, although YdcA possesses a catalytic dyad and other rhomboid protease signature motifs, it was found proteolytically inactive with model substrates *in vitro* (Lemberg et al EMBO J 2005), and it lacks any extramembrane domains. To test whether it can nevertheless behave as a pseudoprotease in relation to MgtE, we created the following strains of *B. subtilis* (*ywld* mutant background) and carried out manganese shock experiments with them.

Wild type; $\Delta yqgP$; $\Delta yqgP\Delta ydcA$; $\Delta yqgP\Delta ydcA+yqgP$

The corresponding Fig. EV1 and text on page 12 show that deletion of *ydcA* does not modify the phenotypic behaviour of *B. subtilis* regardless of the presence or absence of *yqgP*, indicating that endogenous *ydcA* does not play any role in the proteostasis of MgtE during Mn stress, and it is not the reason for the recovery of the $\Delta yqgP$ strain in time.

Minor comments:

1. I am curious about the recognition of YqgP for MgtE. Using the delta G predictor by von Heine's lab (<http://dgpred.cbr.su.se/index.php?p=home>) it seems the second TM is not so favourable for insertion and I presume that it is driven to insert by TM1, which makes sense given the short loop between TM1 and TM2 segments. Since Mg²⁺ is coordinated directly by TM2 (Nature Communications volume 5, Article number: 5374 (2014)), I guess its possible that MgtE may not form a stable fold with low Mg²⁺ and therefore becomes more susceptible to degradation? I was curious if you had analysed MgtE mutants that abolish Mg²⁺ coordination by TM2?

This is an intriguing proposition. Indeed, Mg²⁺ and Mn²⁺ are coordinated by the periplasmic M2 and M3 sites formed by TM1 and TM2 (Nature Communications volume 5, Article number: 5374 (2014)), and these interactions (or lack thereof) may influence the integration of the mildly hydrophobic TM2 into the membrane. However, we have not done any experiments to test this idea, because structural information guiding the required mutant design is available only for the *Thermus thermophilus* MgtE, which is slightly different from *B.subtilis* MgtE (see below). Any mutagenesis experiments in *Bsub* MgtE based on the structure of *Th* MgtE might be difficult to interpret unambiguously because not all residues forming the mentioned periplasmic metal binding sites are conserved (see below), and may require a large body of work that would extend beyond the scope of the present manuscript.

Q5SMG8 MGTE_THET8	1	-----MEEKLAVSLQEALQEGDTRALREVLEEIHPQDLLALWDELKGEHRYVVLTLPLKA	55
034442 MGTE_BACSU	1	MVQNMTYDELILRLIIILLRDGKIRDFRSVIDELQPYDMAFIPKEMPEKHRARYLSYLTVD	60
		: * : : * : : * : : * : : * : : * : : * : : * : : * : : * : : * : : * : : *	
Q5SMG8 MGTE_THET8	56	KAAEVLSHLSPREEQAEYLKTLPPWRRLREILEELSLLDLDALQAVRKEDPAYFQRLKDLL	115
034442 MGTE_BACSU	61	DIIDMIGELELEREFQLVVLNKGKTKATLAMNKMDNDLAQLLEEM--DEELKEQLLSSM	117
		. : : : : . * * * : : : : : : : : : : * : : * : : * : : * : : *	
Q5SMG8 MGTE_THET8	116	DPRTRAEVEALARYEEDEAGGLMTPPEYVAVREGMTVEEVLRFRRRAAPDAETIYYIYVVD	175
034442 MGTE_BACSU	118	EASESKAVQLLMNYPADSAGRMMTNRYVWI PQHYTVKDAVVKLKSF AEIAESINLYVIN	177
		: * : * : * * : : * : : * : : * : : * : : * : : * : : * : : * : : * : : *	
Q5SMG8 MGTE_THET8	176	EKGRLLKGVLSLRDLIVADPRTTRVAEIMNPKVVYVRTDTQEEVARLMADYDFTVLPVVDE	235
034442 MGTE_BACSU	178	ESKQLVGVLSYRDLLILGEPEEKVQDLMFTRVISAALQDEEVARLIERYDFLAI PVVEE	237
		* : * * * * * : : : : * : : * : : * : : * : : * : : * : : * : : * : : * : : *	
Q5SMG8 MGTE_THET8	236	EGRLVGI VTDVLDVLEAEATEDIHKLGAVDVPDLVYSEAGFVALWLARVRWLIVLILIT	295
034442 MGTE_BACSU	238	NNVLVGI VTDVLDVLEAEATEDIHKLGAVDVPDLVYSEAGFVALWLARVRWLIVLILIT	295
		: : * * * * * : : : * * * * * : : * * * * * : : * * * * * : : * * * * * : : *	
		TM1 M2 M3 TM2	
Q5SMG8 MGTE_THET8	296	GMVTSSLLQGFEVLEAVTALAFYV PVLGGTGGNTGNQSATLIRALATRDLDLDRWRV	355
034442 MGTE_BACSU	296	GLISGSLISYFEDALKQVVALAFFMPVSGMTGNTGTQSLAVVIRGLSK EEMNKKTVRL	355
		* : : : * * : * : * : * : * * : : * * * * * : : * * * * * : : * : *	
Q5SMG8 MGTE_THET8	356	FLKEMGVGLLGLTSLFL-LVGKVVWDGHPLLLPVVGVSLVLI VFFANLVGAMLPFLRR	414
034442 MGTE_BACSU	356	IFREFRTSIFIGAVCSVLIAIVSIIWQGNALGFEVVASLFLTLII GTMSGTIIPILHK	415
		: : * : : : : * : * : : : : * : * : * * * : : * * * * * : : * : : * : : *	
Q5SMG8 MGTE_THET8	415	LGVDPALVSNPLVATLSVDTGLLIYLSVARLLEAV	450
034442 MGTE_BACSU	416	LKVDPAIASGPLITTLNDILSLLIYFGIATAFIHSL	451
		* * * * * : * * * * * : * * * * * : * * * * * : * * * * * : * * * * * : *	

Referee #3:

Despite detailed structural insights, very little is known about the cognate substrates and biological functions of rhomboid proteases in bacteria. Two co-submitted manuscripts now fill this gap, one identifying substrates of two rhomboids in the

Gram-negative pathogen *Shigella sonnei* and one revealing a rhomboid function in the Gram-positive model bacterium *Bacillus subtilis*. Both manuscripts present exciting results and are valuable contributions to the field. Since they report strikingly different substrates (orphan single-pass transmembrane proteins versus a multi-pass membrane transporter) and biological functions, the discussions would benefit from cross-referencing and comparing the results in case both papers appear back-to-back.

Began et al. used quantitative proteomics approaches to identify substrates and interactors of the *Bacillus* rhomboid protease YqgP. They found that YqgP cleaves the magnesium transporter MgtE under conditions when it imports toxic ions (low magnesium, high manganese or zinc) and that it interacts with the FtsH protease. Interestingly, YqgP is a metal sensor and binds manganese and zinc ions via its N-terminal domain. Furthermore, YqgP delivers MgtE to FtsH for full degradation. The authors use a wide range of biochemical and biophysical techniques and conclude with interesting evolutionary considerations. I only have a few minor questions and comments.

1. Fig. 1E: Might the strikingly high number of ATPase subunits interacting with YqgP be of physiological significance?

As noted above, we omitted the ATP synthase connection due to perceived space limitation, but now are glad to be able to add it back to the discussion section. See minor point 7 of referee 1.

2. Are there sequence similarities between the cleavage sites in MgtE and TatA?

Unfortunately we do not know the precise cleavage site in MgtE, only the cleavage region. There is no obvious similarity (as can be judged by a simple protein sequence alignment) between *Providencia stuartii* TatA and the YqgP cleavage region of MgtE encompassing its TMD2.

3. In the paragraph describing Fig. 7E (page 15 of the PDF), the authors state that FtsH was not able to act on MgtE at all and refer to lines 25-32. Shouldn't this be lines 1-8? In lines 25-32 both proteases are absent.

We apologize for this confusing misnumbering. All is now duly corrected.

4. Page and/or line numbers would facilitate reviewing.

We apologize for this oversight. I particularly care about including page numbers in all my manuscripts and I am not sure how they can have slipped out in this one. Now all pages and lines are numbered in the revised version.

Accepted

12th December 2019

Thank you for submitting your final revised manuscript for our consideration. I am pleased to inform you that based on the positive re-reviews copied below, we have now accepted it for publication in The EMBO Journal.

REFEREE REPORTS

Referee #1:

The authors have provided a revision that now addresses all of the points raised in my initial review. The study is interesting and important, and the conclusions are well supported by the data.

Referee #2:

I am satisfied by the response from the authors and have no further concerns.

Corresponding Author Name: Kvido Strisovsky

Journal Submitted to: The EMBO Journal

Manuscript Number: 102935